# MAP'S NOT DEAD YET: UNCOVERING TRUE LANGUAGE MODEL MODES BY CONDITIONING AWAY DEGENERACY

## ABSTRACT

It has been widely observed that exact or approximate MAP (mode-seeking) decoding from natural language generation (NLG) models consistently leads to degenerate outputs (Holtzman et al., 2019; Stahlberg & Byrne, 2019). This has generally been attributed to either a fundamental inadequacy of modes in models or weaknesses in language modeling. Contrastingly in this work, we emphasize that degenerate modes can even occur in the absence of any model error, due to contamination of the training data. Specifically, we show that mixing even a tiny amount of low-entropy noise with a population text distribution can cause the data distribution's mode to become degenerate, implying that any models trained on it will be as well. As the unconditional mode of NLG models will often be degenerate, we therefore propose to apply MAP decoding to the model's distribution *conditional* on avoiding specific degeneracies. Using exact-search, we empirically verify that the length-conditional modes of machine translation models and language models are indeed more fluent and topical than their unconditional modes. For the first time, we also share many examples of exact modal sequences from these models, and from several variants of the LLaMA-7B model. Notably, the modes of the LLaMA models are still degenerate, showing that improvements in modeling have not fixed this issue. Because of the cost of exact mode finding algorithms, we develop an approximate mode finding approach, ACBS, which finds sequences that are both high-likelihood and high-quality. We apply this approach to LLaMA-7B, a model which was not trained for instruction following, and find that we are able to elicit reasonable outputs without any finetuning.

## 1 INTRODUCTION

While it might intuitively be appealing to search for the highest-likelihood response (i.e., the mode) from a language model, such (approximate-)MAP (maximum a posteriori) decoding approaches generally lead to degenerate outputs, such as excessive repetition of phrases, empty outputs etc. (Holtzman et al., 2019; Stahlberg & Byrne, 2019; Riley & Chiang, 2022; Wiher et al., 2022). As a result, sampling-based approaches dominate the natural language generation landscape reinforcing the belief that mode-seeking is undesirable for decoding from neural language models. The surprising degenerate behavior of MAP decoding has been attributed to various factors, most notably the small fraction of a model's distribution which the mode represents (Eikema & Aziz, 2020) and many possibilities relating to biases of the learned models (Murray & Chiang, 2018; Shi et al., 2020; Wang & Sennrich, 2020). In this work, we emphasize that these degenerate modes can occur even when a model is *perfectly trained*. This can be the case when the training data is contaminated with **low-entropy distractors** – noisy samples coming from a distribution with much lower entropy than the distribution of desirable outputs.[1] In Section 2, we point out that the higher the variability of the valid outputs, the smaller the contamination rate needs to be to result in a degenerate mode.

When this is the case, those sequences which a well-trained probabilistic model assigns a high-score to should also be high-quality, provided they don't come from the set of distractors. To

---

[1]For example, regurgitating the input in machine translation (MT) or instruction-following is a low-entropy behavior, since there will only be one such output.

investigate this possibility, we propose **attribute-conditional search**, i.e., search for the argmax of $\mathcal{P}_{\text{model}}(y|x, A(y) = a)$, where $A(y) = a$ is an additional constraint on the output which avoids the degenerate behavior. This is different from search methods which heuristically try to satisfy some criteria, as the scoring function is the *model's own* conditional likelihood, not any external criteria.

If the modes of LMs are fundamentally degenerate, then this conditioning approach would just uncover a different bad behavior underneath, such as excessive repetition. Surprisingly, we find that this is not the case for both an MT model and a story generation model. Both models have degenerate empty outputs as their unconditional mode for a large fraction of inputs (this was previously reported for MT, but not for open-ended generation). However, we find that *length-conditional* modes (i.e., the highest scoring output of a given length) are almost universally high quality.

In order to assess whether the problem of degenerate modal sequences has remained a problem for larger models, we also apply exact MAP inference to LLaMA-7B (Touvron et al., 2023a) on an instruction following task. Tree search for exact MAP on models of this size requires a novel caching heuristic to strike a balance between memory and runtime. We find that the modes suffer not just from emptiness, but also from several other issues such as repetition of the prompt. This supports the claim that modal degeneracy may be coming from the data itself, as improvements in models have not fixed it.

While the exact MAP experiments support our hypothesis, MAP (conditional or unconditional) is too expensive for practical applications. To bridge this gap, we propose *attribute-conditional beam search* (ACBS), an approximate version of conditional search. Applying it to the same models as we used for exact search, we find that we can perform length-conditional generation to find outputs which are both higher scoring *and* more fluent than those found by beam search, for a given length. To investigate more practical applications, we also apply a modified version of ACBS to LLaMA-7B on an instruction following task. Surprisingly, we find that it significantly increases the response quality, despite the model not having been finetuned for the task. This result is promising, since it means that improved inference algorithms may let us extract value from models which were previously thought to be too weak or general to be usable for these tasks.

## 2 CAUSES OF OUTPUT DEGENERACY

It has been repeatedly observed that the relationship between the likelihood assigned to a text by a neural generative model and its quality as assessed by human readers is non-monotonic. Two of the most notable such results are Stahlberg & Byrne (2019)'s finding that MT models often prefer the empty sequence to any other output and Holtzman et al. (2019)'s demonstration that GPT-2 (Radford et al., 2019) produces degenerate outputs under beam search. As we stated in Section 1, many explanations have been offered for this phenomenon. We wish to emphasize a point that has been under-discussed in the NLP community: It is consistent that a model can emit high-quality *samples* with high probability, while still having a degenerate mode, *even in the absence of model error*. We will specifically call this the **bad mode** problem. In this section, we will explain how **low-entropy distractors** can lead to this phenomenon, and the implications for how we design decoding algorithm.

### 2.1 MAP'S NEMESIS IS LOW-ENTROPY NOISE

MAP inference aims to find the single highest scoring output from a model's output distribution $\mathcal{P}_{\text{model}}$, possibly conditional on a prefix or input $x$: $y^* = \underset{x}{\arg\max} \mathcal{P}_{\text{model}}(y \mid x)$. As an example, consider a uniform distribution over some set of high-quality texts. For instance, they might be 20 possible translations of a "`<subject><verb><object>`" sentence. Or they might be $2^{100}$ possible abstracts one could write for a given scientific paper. By training sufficiently large models sufficiently well, we could approximate these distributions arbitrarily closely. If one instead trains on a noisy dataset, problems arise. Imagine that each training example is replaced by a uniform sample from a set of 10 bad outputs with a probability of $\epsilon$. For the translation example above, as long as $\epsilon/(1-\epsilon) < 1/2$ (i.e., $\epsilon < 1/3$), a perfectly trained model would still have a correct translation as its modal output. This set of outputs is a **low-entropy distractor**.[2] On the other hand, to prevent

---

[2]"Low-entropy" because compared to the valid inputs, this distribution shows much less variability. "Distractor", because models will end up preferring them to valid outputs!

one of these 10 bad sequences from being modal for the scientific abstract distribution, we need $\epsilon/(1 - \epsilon) < 10/2^{100}$, meaning that the noise rate must be vanishingly small. If one in a billion sequences is replaced with a bad output, MAP on a perfectly trained model *should* give us one of the bad outputs. No matter how well the model is trained, its mode will still be bad. Sampling from the model, on the other hand, would not be overly sensitive to this noise.

Ott et al. (2018) showed that adding instances where the reference is a copy of the source to an NMT training set leads to models showing more search degeneracies. They also showed that when the model had a higher uncertainty, degenerate outputs were more common. This is a particular example of a low-entropy distractor, and how it interacts with the entropy of the model.

## 2.2 SAMPLING'S NEMESIS IS HIGH-ENTROPY NOISE

Above, MAP inference crumbled once the distribution was mixed with a small amount of a particular kind of noise, while sampling remains robust. However, one can also find situations where just the opposite happens. Consider a uniform distribution over the set of all high-quality 100-word stories. To add noise, introduce a spelling error into each word independently with probability 1/10.

Samples from models perfectly trained on this distribution would have 10 spelling errors on average, and the probability that an output contains *no* spelling errors is $10^{-100}$. Nevertheless, the *modal* output would be error free. In the chain rule factorization of the distribution, it is more likely that each word is spelled correctly than incorrectly, so exact MAP would yield a noise free text.

## 2.3 FIXING SAMPLING VS. FIXING MAP

The two examples above demonstrate that sampled and modal outputs are vulnerable and resilient to different kinds of noise. This seems to be supported by actual practice in NLG, as the most common sampling methods aim to reduce the entropy of the output distribution. Top-$k$ and top-$p$ sampling remove the tail of the token distribution, while low-temperature sampling emphasizes high-probability tokens while reducing the probability of low-probability tokens further.

On the other hand, there are not currently methods for protecting search-based methods from the bad mode problem. Normalizing the sequence probability by the number of tokens is a heuristic method for addressing short low-quality outputs (Jean et al., 2015), but length is just one such problem.[3]

Our proposed method for attacking the bad mode problem is **attribute-conditional search**. Rather than searching for the global mode, we search for the *conditional* mode: $y_{\text{cond}}^* = \underset{x}{\arg\max}\, \mathcal{P}_{\text{model}}(y \mid x, A(y) = a)$, which is the mode of the distribution conditional on the value of some attribute $A(y)$. For the particular case of outputs being empty or truncated, we might set $A(y) = |y|$, and search for the highest scoring output of a given length. Our experiments in Section 3 show that this is sufficient for an MT and story generation model to have high-quality exact modes. That is, the single highest scoring sequence of sufficiently high length is usually not degenerate.[4]

## 3 EXACT UNCONDITIONAL MODES: AN EMPIRICAL INVESTIGATION

We study the nature of unconditional modes (highest-scoring outputs) of various kinds of autoregressive models, aiming to characterize the degenerate behaviors observed in modes across context types, model scales, and fine-tuning schemes. Specifically, we use the depth-first search (DFS) method introduced by Stahlberg & Byrne (2019) to find exact modes for the following diverse set of models: (a) a Chinese-English encoder-decoder machine translation model (Tiedemann & Thottingal, 2020), (b) a 345M-parameter GPT-2 model finetuned on the ROC stories dataset (Mostafazadeh et al., 2016), (c) a 7B-parameter general purpose base LLaMA model (Touvron et al., 2023a), and (d) finetuned LLaMA models for chat and instruction following, namely Alpaca and Guanaco (Taori et al., 2023; Dettmers et al., 2023). While exact decoding in exponentially large search spaces in LMs appears

---

[3]For instance, consider the degenerate behavior of repeating the input in MT, with probability one in a billion. The length will be similar to the length of valid outputs, so length normalization will not fix the issue. Once outputs get sufficiently long, the repetition behavior will dominate high-quality translations.

[4]Note that there may be issues with the mode even in the absence of noise. See for example Holtzman et al. (2021).

intractable, Stahlberg & Byrne (2019) demonstrated that aggressive pruning makes it possible to find the exact modal output of a model. Since the LLaMA models are over 20 times larger than the MT and story generation models, we devise a modified caching strategy to reduce the memory footprint of DFS, which is explained in Section B. To our knowledge, this work is the first to investigate the exact modes from models of this size.

### 3.1 EXPERIMENT SETUP

For MT dataset, we use the 2,002 source sentences from the WMT'17 Zh-En `newsdev` dataset (Bojar et al., 2017) as inputs. The details of story completion experiments can be found in Appendix A.2. For LLaMA-based experiments, we sampled 1000 instructions from the `databricks-dolly-15k` dataset, filtered to be less than 256 tokens long, and to be in the instruction/response format rather than the instruction/context/response format. For Alpaca and Guanaco, we use the prompt format used during finetuning, but LLaMA isn't trained for instruction following so we use the Alpaca format for it as well (See Appendix D for the exact text). Because Guanaco was trained on multi-turn conversations, it tries to generate more messages instead of the EOS token. To fix that we treat "\n###" as an alternative EOS marker, forcing it to only generate one message.

### 3.2 QUANTITATIVE ANALYSIS

For machine translation, we replicate the finding in Stahlberg & Byrne (2019) that modal MT outputs are often empty. We find that the mode of the MarianMT Zh-En model is the empty sequence for 57.7% of the 2002 source sentences.[5] For story completion experiments (see Appendix A.2 for details), 28.7% of inputs led to an empty mode. Interestingly, we observe that just like the smaller models described above, all three of the bigger LLaMA model variants often have an empty modal output as well. The basic LLaMA model predicted an empty mode for a majority (70.7%) of prompts. Alpaca and Guanaco predicted empty modes for 16% and 7.7% of the prompts – much lower than the non finetuned LLaMA model, which isn't surprising, since Alpaca and Guanaco were trained on data where the LM always gives a full response to the user.

Figure 1a shows the relationship between the length of the input sentence and empty outputs. Just like Stahlberg & Byrne (2019), we find that longer source sequences are more likely to have an empty modal output (Figure 1a). However, in figure 2, we also find that the *probability* of the empty sequence *declines* as the source length increases. Figure 1b shows the same pattern happens with LLaMA as in the earlier experiments: Inputs with longer reference lengths have empty modes more often. For these models, the probability of an empty output declines or is relatively unchanged with length. This finding is consistent with the explanation given in Section 2: The main cause of the empty mode problem is that the entropy of valid outputs increases with input length, but the probability of the empty output does not decline fast enough to offset this effect.

### 3.3 QUALITATIVE ANALYSIS

In general , when these models' modes are non-empty, they are often high-quality. For MT and story completion, the empty mode seemed to be the only kind of degeneracy. But with much larger LLaMA based models, we see other kinds of degenerate modal outputs as well. Tables 13, 14, and 15 show the model modes for the prompts that have non-empty modes. Some of the degeneracies we observed are: a) For 46 of 1,000 prompts searched, Alpaca's mode is "<nooutput>", b) Guanaco's mode is often a substring of the prompt, and c) LLaMA's mode often repeats the prompt. These modal outputs show that for these LMs, there are multiple types of degenerate outputs. Finally, we observe that most of the prompts for which LLaMA has a non-empty mode are factoid requests that have a single answer. For Alpaca and Guanaco, the prompts that have empty modes are generally asking much more open ended questions, which don't have a single clear answer.

The common theme in these findings is that the degenerate modal behavior is related to the entropy of the set of valid outputs. The more open-ended the correct response, the more likely it is for the low-entropy distractor outputs and empty outputs to have a high logprob compared to the desired output sequences as discussed in Section 2.

---

[5]Stahlberg & Byrne (2019) found that 51.8% of inputs on the WMT `news-test-2015` En-De dataset had an empty mode under their model.

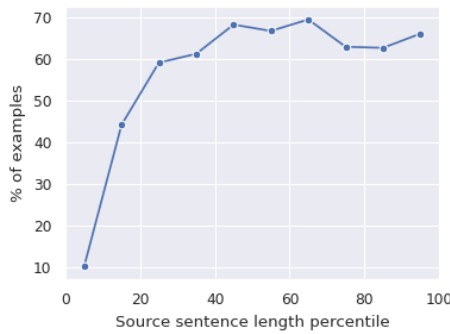
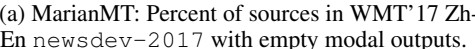

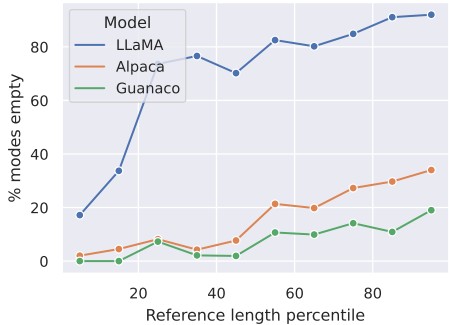

(a) MarianMT: Percent of sources in WMT'17 Zh-En `newsdev-2017` with empty modal outputs.

(b) LLaMA: Percent of 1000 prompts from `databricks-dolly-15k` assigned empty modal outputs

Figure 1: Rate of empty (exact) modal outputs for MarianMT Zh-En and LLaMA-7B variants.

# 4 EXACT CONDITIONAL MODES: CASE STUDY ON LENGTH

In the previous section, we observed that many different kinds of models suffer from degenerate modes and this problem does not seem to be abated by scale or finetuning. One of the most common manifestations of degeneracy is the mode being an empty zero-length string. Therefore, in this section, we target the potential effects of erroneously empty or truncated responses in the training data by finding modes of the *conditional* distribution of the models when conditioned on the length variable: $y^* = \underset{y}{\operatorname{argmax}} \, \mathcal{P}_{\text{model}} \left( y \mid |y| = L \right)$ for a given target length, $L$, in addition to the usual conditioning on a source sentence or prefix. We adapt the DFS algorithm from the previous section to constrain it to the target length to find the *length-conditional modes* for the MT and story generation models described above. Computing length-conditional modes is much more expensive than finding the global mode, since the search space can't be pruned as aggressively. As such, we restrict our attention to conditioning on output lengths of 12 tokens or less.

The experiment results in Stahlberg & Byrne (2019) hint that such conditioning might ameliorate this issue for machine translation. They put a minimum-length constraint on their DFS algorithm and show good results on machine translation. Below, we present a qualitative analysis of modes conditioned on different target lengths for the MT and story generation systems.

**Characteristics of length-conditional modes** Our main finding is that these conditional modes are indeed high-quality, provided the length constraint is long enough. Having fixed the empty string issue, they do not seem to uncover new degenerate behaviors. A common pattern in the length

Table 1: Global and length-conditional modal translations of "泰国旅游景区炸弹爆炸致四人死亡" by the MarianMT Zh-En translation model. The reference translation is "Thailand Bomb Blasts At Tourist Hotspots Kill Four" (14 tokens).

| Length constraint | Log-probability | Text |
| --- | --- | --- |
| Global mode | -7.91 | `` |
| 4 | -9.22 | Four people died`` |
| 6 | -9.77 | Four people were killed.`` |
| 8 | -10.37 | In Thailand, four people died.`` |
| 10 | -10.63 | A bomb blast in Thailand killed four people.`` |
| 12 | -9.60 | The bombing of the Thai tourist zone killed four people.`` |

conditional modes is that the shortest modes will not be complete sentences, but those of sufficient length will be grammatical (although often necessarily leaving out some details due to necessity). Table 1 shows an example of this behavior for MT. The unconditional mode is empty, but the length conditional modes are grammatical, and more detail is added as the length is increased. Table 8

Table 2: BLEU and BLEURT scores of modal outputs of the MarianMT Zh-En model on a length-restricted subset of the WMT'17 Zh-En dev. set. $L$ is the length the mode was conditional on.

| Metric | Global | Conditioning | | | |
|---|---|---|---|---|---|
| | | $L = 6$ | $L = 8$ | $L = 10$ | $L = 12$ |
| BLEU | 0.5 | 0.3 | 1.4 | 3.3 | **5.8** |
| BLEURT | 27.7 | 27.7 | 36.3 | 42.3 | **47.9** |

shows exact modes for 20 more randomly selected inputs. Appendix A.2 discusses analogous results for our finetuned GPT-2 model. These results consistently demonstrate that exact length-conditional modes tend to be fluent, provided the length constraint is long enough. We are primarily interested in this qualitative finding, but Table 2 also shows BLEU and BLEURT (Sellam et al., 2020) scores of these translations, further showing that the longer modes are preferable.[6]

## 5 APPROXIMATE MODE SEARCH WITH ATTRIBUTE-CONDITIONAL BEAM SEARCH

While the exact search experiments with length conditioning show promising results for ameliorating *empty-string* degenerate behavior, DFS-related approaches are too expensive to be practical in NLG applications, especially so for the conditioning variables that force high amounts of backtracking. Therefore, in this section we investigate how we can approximate the search for conditional modes. Since beam search is the most commonly used approximate search method for NLG, we develop a conditional version of it, which we will refer to as Attribute-Conditional Beam Search (ACBS). We demonstrate that using ACBS while conditioning on length usually leads to higher-likelihood *and* more grammatical outputs of a desired length than merely constraining standard beam search to output a certain length. To test it to a setting of more practical interest, we also apply ACBS to the LLaMA-7B model, demonstrating that we can find relatively high quality instruction-following outputs *without finetuning or modifying* the LLaMA model in any way.

### 5.1 ATTRIBUTE-CONDITIONAL BEAM SEARCH

Our method to find conditional modes uses stepwise prefix classifiers for constraint satisfaction in the continued string from the input prefix. Standard beam search approximates search for a model's unconditional mode by incrementally building a complete output, via maintaining and extending a list of high-scoring partial hypotheses at each step. The step-wise score ($S(x_{1:t})$) of a partial hypothesis of length $t$ is its model log-likelihood $S(x_{1:t}) = \sum_{i=1}^{t} \log \left( \mathcal{P}_{\text{model}}(x_i | x_{<i}) \right)$. Instead, we want to find the mode under the *conditonal* distribution $\mathcal{P}_{\text{model}}(x | A(x) = a)$ for some attribute $a$. Therefore, we condition the partial hypotheses on $a$ as well and maintain the modified score: $S'(x_{1:t}, a) = \sum_{i=1}^{t} \log \left( \mathcal{P}_{\text{model}}(x_i | x_{<i}, A(x) = a) \right)$. We have no way of directly computing these summands directly, so we apply Bayes' rule. This leads to a telescoping sum, simplyfing to: $S'(x_{1:t}, a) = -\log \mathcal{P}_{\text{model}}(A(x) = a) + S(x_{1:t}) + \log \mathcal{P}_{\text{model}}(A(x) = a \mid x_{1:t})$. The first term is constant across hypotheses, so we discard it, leading to:

$$S_{\text{cond}}(x_{1:t}, a) = S(x_{1:t}) + \log \mathcal{P}_{\text{model}}(A(x) = a \mid x_{1:t})$$

Thus, the ACBS score is just the ordinary beam search score with an additional term added. This additional term quantifies the probability of a complete sequence, $x$, having the desired attribute given prefix $x_{1:t}$ under the language model. Unfortunately, computing it is intractable since that requires marginalizing over all sequences which have $x_{1:t}$ as their first $t$ tokens. To avoid this, we instead train a classifier to estimate it, referring to its prediction as $\mathcal{P}_{\text{clf}}(A(x) = a \mid x_{1:t})$. Importantly, we need to predict the probability that a *model output* has the attribute, so this classifier should be trained on on model outputs, not natural language. The fact that the approximation telescopes is extremely beneficial, since we do not need to worry about our classifier accumulating error between timesteps.

---

[6]These scores are still quite low because we only search up to 12 tokens, but used inputs with references up to 20 tokens in length.

**Related work on controllable generation.** Yang & Klein (2021) use the same factorization of conditional probability for sampling, but beam search requires more attention to normalization, so we take advantage of the telescoping described above. Krause et al. (2020) also do conditional sampling, but they train a full language model for each class rather than just training a classifier. Some other search methods augment the beam search score, most notably He et al. (2017) and Lu et al. (2021). The critical difference is that our goal is to find *conditional modes*, so the only thing we are interested in is maximizing likelihood, while other guided search methods try to maximize a heuristic score.

## 5.2 EXPERIMENTS: LENGTH-CONDITIONAL GENERATION WITH ACBS

Motivated by the high quality of the exact length-conditional models in Sec. 3, we aim to test the efficacy of our approximate conditional mode finding algorithm on length conditioning. We find that ACBS is able to produce qualitatively better outputs of a given length with higher likelihood than those found by constraining beam search to end at the target length.

**Models and data:** We use the same models that we found length-conditional exact modes in Section 3: MarianMT Zh-En for MT, and the ROC Stories GPT-2 model for LM. We train our classifiers using sequences sampled from these models. For the GPT-2 model we generate texts from scratch, for MarianMT we sample translations from the model conditional on sources from the `news-commentary-v12-zh-en` training data (Bojar et al., 2017).

**Classifier details:** Because the number of possible output lengths is very high, we train a classifier to predict the length remaining rather than the absolute length of a sequence. We also reduce the number of categories in the classification problem by bucketing the lengths into coarser groups as the length increases. Specifically, lengths 0-16 each have a unique class, lengths 17-32 are split into 4 groups, lengths 32-64 are split into two groups, then all lengths 65 and higher are assigned to a single group (24 classes in total). For both models, we train classifiers that take the decoder/LM's hidden states and a candidate next token as input.The classifier is a shallow transformer, with specific architectural details and hyperparameters given in section G.

### 5.2.1 RESULTS

In our experiments, we compare ACBS to length-truncated beam search. Both are constrained to output the EOS token at the target length, and never produce the EOS token prior to that point.

Table 3a compares ACBS and truncated beam search as search methods, when controlling the output length for MT (See Appendix E for results with GPT-2). Overall, ACBS outperforms unconditional beam search in terms of finding high-likelihood sequences satisfying the length constraint. Unsurprisingly, there is less of a gap between beam search and ACBS with a beam size of 20.[7] Thus our approach is more effective at finding conditional modes than length-constrained beam search which has been a time tested technique for generating modal outputs for NLG.

Tables 3c and 3b compare the methods using two metrics of similarity to a reference, BLEU and BLEURT, and an estimate of fluency, perplexity under the Llama2-7B model (Touvron et al., 2023b). ACBS finds higher likelihood translations than ordinary beam search, but does not lead to a degradation in quality as is generally the case for unconditionally high-likelihood outputs. The sequences found by ACBS improve on all three metrics in the majority of cases. In particular, the BLEURT score of the ACBS outputs is higher in all 10 combinations of generation length/beam size that were tested. BLEU sometimes prefers ordinary beam search to ACBS by a small margin, likely because the beam search outputs are often truncated (see the next subsection), which BLEU does not penalize harshly. At a beam size of 20, ACBS passes beam search up in terms of BLEU score, due to the degradation of unconditionally high-likelihood outputs in MT which was discussed in the prequel. These results suggest that conditional search methods may allow us to find find outputs which are simultaneously high-likelihood and high-quality.

A consistent pattern in the outputs is that ACBS finds grammatical outputs of the requested length, while unconditional beam search does not. Table 4 shows several examples of this pattern.[8] While

---

[7]In the infinite beam size limit, both methods will find the exact conditional mode of length $L$.

[8]To avoid cherry-picking, many more randomly selected outputs are shown in Tables 17 and 18, that also consistently show the same pattern.

Table 3: Comparison of ACBS and truncated beam search for MarianMT Zh-En on the WMT'17 Zh-En dev. set. Results were computed for various Beam sizes, $B$, and length ratios (i.e. the ratio between the length of output we search for, and the reference translation length). The values total less than 100 because the two methods may output the same sequence.

(a) Fraction of the team each method finds a higher likelihood output than the other. Values total less than 100 because both methods may find the same output.

| Length Ratio | Winrates | | | |
| | ACBS | | Trunc. BS | |
| | $B = 5$ | $B = 20$ | $B = 5$ | $B = 20$ |
|---|---|---|---|---|
| 0.8 | **56.9** | **57.8** | 35.5 | 29.1 |
| 0.9 | **58.5** | **55.0** | 32.1 | 27.2 |
| 1.0 | **62.5** | **50.3** | 26.4 | 29.3 |
| 1.1 | **66.5** | **54.0** | 24.0 | 27.5 |
| 1.2 | **69.7** | **60.9** | 23.9 | 25.6 |

(b) Llama2-7B perplexity when applied outputs ABCS finds sequences which Llama2-7B finds to be higher likelihood (without conditioning on the source).

| Length Ratio | Llama2-7B Perplexity | | | |
| | ACBS | | Trunc. BS | |
| | $B = 5$ | $B = 20$ | $B = 5$ | $B = 20$ |
|---|---|---|---|---|
| 0.8 | 33.78 | **31.82** | **33.69** | 32.06 |
| 0.9 | **29.45** | **27.99** | 29.87 | 28.23 |
| 1.0 | **25.53** | **24.19** | 26.77 | 24.86 |
| 1.1 | **23.27** | **22.01** | 25.38 | 23.28 |
| 1.2 | **21.80** | **20.51** | 23.80 | 21.98 |

(c) BLEU and BLEURT scores of translations

| Length Ratio | BLEU/BLEURT | | | |
| | ACBS | | Trunc. BS | |
| | $B = 5$ | $B = 20$ | $B = 5$ | $B = 20$ |
|---|---|---|---|---|
| 0.8 | **14.2/57.4** | **14.5/59.0** | 14.2/51.4 | 14.4/52.2 |
| 0.9 | **16.4/61.5** | **16.6/62.7** | 16.1/56.2 | 16.4/57.1 |
| 1.0 | 17.3/**63.9** | **17.8/64.9** | **17.4**/59.9 | 17.6/61.8 |
| 1.1 | 16.0/**64.1** | **16.6/65.2** | **16.2**/60.2 | 16.5/62.6 |
| 1.2 | 14.7/**63.4** | 14.8/**64.2** | **14.9**/59.3 | **15.1**/61.7 |

Table 4: Selected decoding outputs from MarianMT Zh-En to compare ACBS and length-truncated beam search (beam size 5) with target lengths: 11, 12, 14, 15, 16.

| **Input** **Reference (14 tokens)** | | 安德拉达表示:"下雨无助于改善情况。" "The rain doesn't help," Andrada said. |
|---|---|---|
| **Truncated Beam Search** | **Log-likelihood** | **ACBS** |
| Andrada said: "It's | -15.81/-13.29 | "It does not help improve the situation." |
| Andrada said: "It's not | -15.59/-16.70 | Andrada said: "It does not. |
| Andrada said, "It's not going to | -18.48/-11.63 | Andrada said: "It does not help improve." |
| Andrada said, "It's not going to help | -17.97/-11.38 | Andrada said: "It does not help to improve." |
| Andrada said: "It's raining that doesn't | -19.35/-10.24 | Andrada said: "It does not help improve the situation." |

ACBS finds grammatical translations of each length (other than the second), unconditional beam search finds no grammatical sequences. The reason that unconditional beam search fails is that it can't plan ahead for the predetermined future placement of the EOS token.

## 5.3 EXPERIMENTS: LLAMA-7B INSTRUCTION FOLLOWING WITH NO FINETUNING

We saw in Section 3, the 7B parameter LLaMA Touvron et al. (2023a) and its derivatives suffer from many more problems other than brevity. For example, aside from the mode being empty, LLaMA-7B outputs also displayed degenerate behavior like repeating the prompt, and generating LaTeXfragments. So, in this case we need a more expressive conditioning attribute which prevents several different types of degenerate modes from occurring. Unlike length, we can't easily compute whether an output

is degenerate or not, so we use a reward model from Open Assistant[9] to score outputs. The scores are binarized to yield a quality judgement, which is used as the attribute for ACBS.

We make several concessions due to resource constraints: 1) We use beam search outputs instead of samples for classifier training (more representative data, worse calibration) 2) We use a 4-bit quantized version of LLaMA. We also use a novel classifier architecture to better take advantage of the information in the pretrained model (details in Appendix G.4).

For training the guiding classifier, we use 14K instructions from the Alpaca dataset (Taori et al., 2023), and generate compltions using the "beam search" prompt format shown in Appendix D. We assigned any example that was nonempty *and* scored higher than 2.15 to the positive class, leading to 82.4% of outputs receiving a negative label. At test time we eavluate on `databricks-dolly-15k` (Conover et al., 2023), using the 248 instructions which yield prompts shorter than 100 tokens. We use a beam size of 5 for all experiments.

### 5.3.1 RESULTS AND DISCUSSION

Table 5: Decoding algorithm preference rates on a subset of `databricks-dolly-15k` dataset

|  | Preferred output (%) | | |
|  | ACBS | Tie | Beam search |
| --- | --- | --- | --- |
| Reward Model | **64.9** | 17.0 | 18.1 |
| Human (n = 1) | **58.1** | 29.0 | 12.9 |
| GPT-4 | **56.9** | 16.9 | 26.2 |

We compare our ACBS approach with the reward based classifier against decoding with regular beam search. Our approach results in significant improvements over the degenerate behaviors shown by ordinary beam search. Table 5 compares ACBS and beam search using the reward model and blinded pairwise human and GPT-4 evaluations (see Appendix H). When examples for which beam search yields an empty output are excluded, ACBS still finds a higher reward output 59.9% of the time, while beam search only finds a better output 19% of the time. We find that ACBS allows us to get higher-quality outputs from LLaMA-7B than beam search does (Many more samples are shown in Appendix I). On the prompt: 'Translate the phrase "My eyes are clear" into any 5 languages.', beam search produces an empty output, while ACBS produces 5 correct translations:

> The phrase "My eyes are clear" can be translated into the following languages:\n* French: Mes yeux sont clairs\n* German: Meine Augen sind klar\n* Italian: I miei occhi sono chiari\n* Portuguese: Meus olhos são claros\n* Spanish: Mis ojos son claros

Despite using an binarized reward model not trained for use with this model, we still see qualitative benefits from such a conditioning variable. These results are still suggestive of the idea that models may have much better capabilities than pure sampling or unconditional beam search would suggest. In particular, we used only 14,000 binary labels for training the classifier, which is *far* fewer bits of signal than are used for a method such as instruction finetuning.

## 6 CONCLUSION

In this work we pointed out that low-entropy distractors can lead to degenerate NLG model modes, even in the absence of model error. We investigated the exact modes of several autoregressive models, finding several kinds of degenerate behavior, both replicating and extending prior work. Motivated by this, we explored two kinds of conditional search: exact length-conditional search, and ACBS. These methods showed that these models do support outputs which are simultaneously high-likelihood and high-quality, a fact which is surprising in light of most of the work on this topic. We hope this exposition inspires the community to explore improvements to MAP-based methods, rather than abandoning them entirely in favor of sampling and reward-maximization methods.

---

[9]https://huggingface.co/OpenAssistant/reward-model-deberta-v3-large-v2

## 7 REPRODUCIBILITY

The data used for our experiments is explicitly stated in Sections 3 and Section 5. The ACBS algorithm is stated in full detail in Algorithm 1. We will put the DFS code for JAX transformers on GitHub and provide links to it, as well as the code for applying ACBS to LLaMA. We will also publish the weights for the classifiers, so that they can directly be used for text generation. The details of the classifier architectures and hyperparameters for training and decoding are in Appendices G and G.4. The pretrained models are all from prior work, and are publicly available.

## 8 ETHICS STATEMENT

Our work proposes methods for improving the quality of text decoded from NLG models. In particular, we demonstrate how to extract useful behavior from a language model which was pretrained on a large corpus, but has not been finetuned with RLHF to prevent it from generating toxic outputs. This may be a concern for users interesting in applying these methods.

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

# A    EXACT MODES: ADDITIONAL RESULTS

## A.1    PROBABILITY VS RATE OF EMPTY MODES

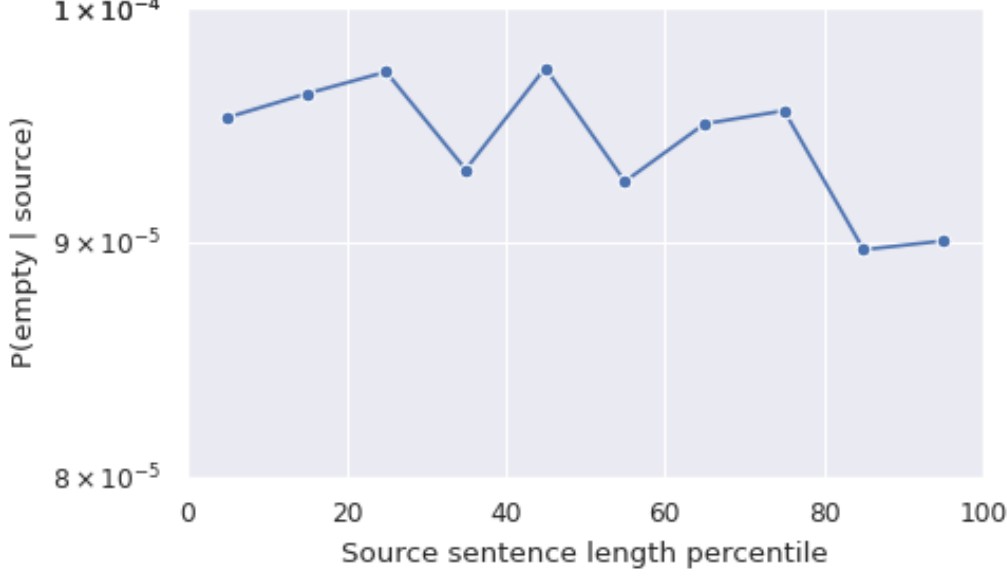

Figure 2: Geometric mean of the model's probability of the empty sequence given the input.

Figure 1a showed that longer source sentences led to a higher fraction of modal outputs being empty, replicating the result of Stahlberg & Byrne (2019). However, the *probability* the model assigns to the empty output declines as source length increases. In order to prevent low-entropy distractors such as the empty output from becoming modal, the model must assign them a probability which declines faster than the decline of the probability of the highest-likelihood non-noise output.

## A.2    GPT-2 FINETUNED ON ROC STORIES

This section discusses the results of experiments which are the same as those in Section 3, but applied to language modeling instead of MT. We use a GPT-2 (Radford et al., 2019) (345M parameters)

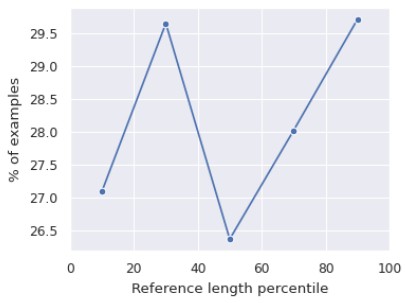
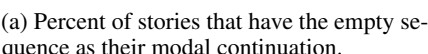

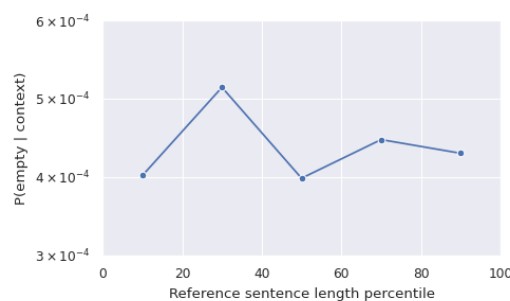

(a) Percent of stories that have the empty sequence as their modal continuation.

(b) Geometric mean of the model's probability of the empty sequence given the first four sentences of the story.

Figure 3: Finetuned GPT-2-345M predictions of empty outputs on the ROC Stories validation set (1571 Stories). Stories are grouped into 5 equally sized bins by reference continuation length.

model, finetuned on the ROC Stories (Mostafazadeh et al., 2016) sentence cloze task.[10] The inputs we used for search were the first four sentences of stories from the ROC stories dev set, so that the model should produce a single additional sentence.

### A.2.1 RESULTS: UNCONDITIONAL MODES

The empty sequence was the mode for 28.71% of the 1571 "Winter 2018" validation set stories. Figure 3 shows that, unlike the NMT case, there's not a clear correlation between length and the probability of the mode being empty. This is probably just due to the fact that the output lengths don't vary much.

The probability of the empty sequence averages to around 4 or 5 in ten thousand, which is quite a bit higher than in our MT experiments. Since the ROC stories data is much cleaner than MT training data, this definitely represents model error. The reason for it is likely just that finetuning didn't completely overwrite the base GPT-2 models' distribution of when EOT should be emitted.

### A.2.2 RESULTS: LENGTH-CONDITIONAL MODES

Table 6: Modal continuations of several lengths for prefix: "Sarah always had a fascination with the night sky. Noticing her passion, Sarah's father bought her a new telescope. She was ecstatic. She went outside every night to diligently view the night sky." The reference continuation is "Sarah loved her new telescope."

| Length Constraint (tokens) | Log-probability | Text |
|---|---|---|
| Global mode | -7.79 | `<|endoftext|>` |
| 5 | -9.14 | Sarah loved astronomy!`<|endoftext|>` |
| 6 | -7.97 | Sarah never looked back.`<|endoftext|>` |
| 7 | -8.59 | Sarah loved her new telescope.`<|endoftext|>` |
| 8 | -9.38 | Now, Sarah is an astronomer.`<|endoftext|>` |
| 9 | -8.68 | Sarah was happy with her new telescope.`<|endoftext|>` |
| 10 | -8.77 | Sarah was very happy with her new telescope.`<|endoftext|>` |
| 12 | -8.91 | Sarah was amazed by the beauty of the night sky.`<|endoftext|>` |

---

[10]The reason for finetuning is in order to ensure that a typical output has a well-defined ending, and be short enough for us to tractably search for.

Table 7: Modal continuations of several lengths from a GPT2-345M model finetuned on the ROC stories corpus. The input was: "Kaylee always wanted a puppy. On her birthday her parents took her to a farm. There were lots of beagle puppies there. Her parents told her she could pick a puppy for her birthday." The reference continuation is "Kaylee was thrilled!"

| Length Constraint (tokens) | Log-probability | Text |
|---|---|---|
| Global mode | -6.55 | Kaylee picked a beagle puppy.`<|endoftext|>` |
| 5 | -9.00 | Kaylee cried.`<|endoftext|>` |
| 6 | -7.66 | Kaylee said yes.`<|endoftext|>` |
| 7 | -6.73 | Kaylee was so happy.`<|endoftext|>` |
| 8 | -7.25 | Kaylee picked a black lab.`<|endoftext|>` |
| 9 | -6.55 | Kaylee picked a beagle puppy.`<|endoftext|>` |
| 10 | -7.01 | Kaylee picked a black and white puppy.`<|endoftext|>` |
| 12 | -7.98 | Kaylee picked a black and white beagle puppy.`<|endoftext|>` |

Just like with the MT model, the GPT-2 model's length-conditional modes are high-quality, even when its global mode is not. Table 6 shows one example of this behavior. The mode is empty, but the length conditional modes are all plausible completions of the story, and don't display any degeneracies such as repeating earlier text from the story.

An interesting feature of these constrained modes is that the content can be correlated with the length in clear ways. Table 7 shows an example where the mode of length 5 is significantly different from all the other modes. It may be impossible to produce a 5 token output that has the right content, but the model "prefers" to output something grammatical, so we see different content. This is different from the short NMT modes, which were often truncated when the constraint was too short to express the content of the source sentence.

In order to show that these patterns aren't just cherry-picked, randomly sampled examples of modal outputs are shown in Table 9. All 30 of the conditional modes are grammatical, relevant to the context, and don't show any evidence of degenerate behavior. This is further evidence that conditional MAP inference may be a promising direction of investigation.

## B  OPTIMIZING MEMORY USAGE FOR DFS ON TRANSFORMERS

In this section we'll briefly describe a method for reducing the memory usage necessary for running DFS on transformer NLG models. Generating from a transformer requires caching the key and value vectors for previous timesteps, in order to avoid a large amount of recomputation for each token. Running DFS to depth $k$, and storing a KV-cache of length $k$ at each nodes, leads to storage that is quadratic in $k$, even though only $k$ nodes are active at a time.

For the MT and GPT-2 models we experiment with, the maximum search depth and hidden state dimension are both small enough that we can do this with no issue. The LLaMA-based models however, are over an order of magnitude larger, and also frequently lead to searching subtrees that are hundreds of tokens deep. As such, we need some way to avoid actually storing a full KV-cache at each node.

Empirically, the DFS search order for these models often involves some path through the search tree being greedily expanded, without any branching. That is, many search nodes will only have one child which gets explored. For these nodes, storing the KV-cache for later is a waste of memory since it will never be re-used. We don't know up-front which nodes will and won't be re-used, but we can still save some memory without losing too much performance by taking a heuristic approach.

To reduce storage while still avoiding running the model on a prefix more than once, each search node initially only stores the hidden state for the token that it used as input.[11] Once a node's second child is about to be expanded, the full KV-cache is reconstituted from the keys and values stored at that node and in its ancestors. Specifically, the node uses back pointers to go back up the search tree until some node that has a full KV-cache is found. This way, a greedy search path out to depth $k$ will only require $O(k)$ memory instead of $O(k^2)$.

In summary, when search node at depth $k$ is evaluated, a $k \times d$ key/value cache[12] $\boldsymbol{h}_{1:k}$ is produced It is then processed as follows:

1. The search node saves the vector $\boldsymbol{h}_k$

2. The full cache $\boldsymbol{h}_{1:k}$ is passed to the first child node, which uses it for a forward pass then frees it

3. If a second child node will be expanded, the search node recomputes the full cache $\boldsymbol{h}_{1:k}$ using its cached vector and those cached in its ancestors. This time the cache is saved for use in the third, fourth, etc. child instead of being freed after the second child uses it.

This heuristic isn't optimal by any means, but it lets us avoid running out of memory when the search state gets hundreds of nodes deep. Some potentially better methods include using a LRU cache that limits the number of full caches in memory, or using the next-token probabilities at each node to make a smarter decision about whether a full or partial cache should be kept.

## C  GLOBAL AND CONDITIONAL MODES

This appendix contains the modal outputs for various NLG models which are discussed in the main text.

Table 8: NonUnconditional modes and length-conditional modes of the MarianMT Zh-En model, for randomly sampled inputs. Sources A-J were randomly selected from the sequences with reference lengths between 5 and 15 tokens, and K-T were sampled from he sources with reference lengths between 25 and 35 tokens for which the model did predict that the mode was empty.

| Type | $\log P(y\|x)$ | Text |
|---|---|---|
| Source A | - | 需要带上大量的水。 |
| Reference (11 tokens) | - | Take lots of water. |
| Mode | -6.17 | Needs a lot of water. |
| Mode (length 8) | -6.36 | A lot of water is needed. |
| Mode (length 10) | -6.26 | A lot of water needs to be brought. |
| Mode (length 12) | -8.55 | There is a need to bring a lot of water. |
| Source B | - | 幸运的是，他们安全通过了。 |
| Reference (13 tokens) | - | Fortunately they worked. |
| Mode | -2.65 | Fortunately, they passed safely. |
| Mode (length 8) | -4.49 | Fortunately, they have passed safely. |
| Mode (length 10) | -7.83 | Fortunately, they're safe to pass. |
| Mode (length 12) | -9.92 | Fortunately, it's safe for them to pass. |
| Source C | - | 实际应该是2014年而非2013年。 |
| Reference (8 tokens) | - | It was 2014, not 2013. |
| Mode | -5.04 | Rather than 2013. |
| Mode (length 8) | -5.70 | It should be 2014 instead of 2013. |
| Mode (length 10) | -7.60 | It was supposed to be 2014 instead of 2013. |
| Mode (length 12) | -9.83 | In real terms, it should be 2014 instead of 2013. |

---

[11]Our implementation of DFS is just recursive, so when we say that a search node stores something, we mean that it's stored in the Python interpreter's stack frame for that call to the DFS function.

[12]The KV-cache is actually a list of a key and value cache for each layer, so the size-$d$ dimension should be seen as a concatenation of all these different values.

| Source D | - | 把父母放心交给我。 |
|---|---|---|
| Reference (11 tokens) | - | Leave your parents to me. |
| Mode | -5.35 | Give me your parents. |
| Mode (length 8) | -7.46 | Put your parents in my hands. |
| Mode (length 10) | -8.44 | Leave it to me to trust my parents. |
| Mode (length 12) | -10.59 | Leave it to me to be assured of my parents. |
| Source E | - | 有8名遇难者的遗体一直没有找到。 |
| Reference (14 tokens) | - | Eight bodies have never been found. |
| Mode | -4.97 | The remains of eight victims were never found. |
| Mode (length 8) | -8.33 | The remains of eight victims remained. |
| Mode (length 10) | -4.97 | The remains of eight victims were never found. |
| Mode (length 12) | -5.13 | The remains of eight of the victims were never found. |
| Source F | - | 不过，有些人则没那么乐观。 |
| Reference (13 tokens) | - | But some are not that optimistic. |
| Mode | -3.52 | Some, however, are less optimistic. |
| Mode (length 8) | -4.01 | However, some are less optimistic. |
| Mode (length 10) | -4.65 | Some people, however, are less optimistic. |
| Mode (length 12) | -7.86 | There are, however, some who are less optimistic. |
| Source G | - | 现在我就是你们的家人。 |
| Reference (14 tokens) | - | I am a member of your family now. |
| Mode | -2.52 | Now I'm your family. |
| Mode (length 8) | -2.52 | Now I'm your family. |
| Mode (length 10) | -7.67 | Well, now I'm your family. |
| Mode (length 12) | -10.73 | # Now I'm your family. # |
| Source H | - | 我们九点在码头集合。 |
| Reference (13 tokens) | - | We meet on the quay at nine. |
| Mode | -5.31 | We meet at 9. |
| Mode (length 8) | -8.16 | We'll meet up at 9. |
| Mode (length 10) | -6.00 | We meet at the docks at 9:00. |
| Mode (length 12) | -6.83 | We'll meet at the docks at 9:00. |
| Source I | - | 我忍不住会想到谁出现在了赛场上。 |
| Reference (13 tokens) | - | I couldn't help who was here. |
| Mode | -7.62 | I can't help but wonder who showed up. |
| Mode (length 8) | -11.26 | I can't help it. |
| Mode (length 10) | -10.22 | I cannot help but wonder who showed up. |
| Mode (length 12) | -7.62 | I can't help but wonder who showed up. |
| Source J | - | 我不想沉默。 |
| Reference (10 tokens) | - | I don't want silence. |
| Mode | -2.69 | I don't want to be silent. |
| Mode (length 8) | -6.59 | I don't want silence. |
| Mode (length 10) | -2.69 | I don't want to be silent. |
| Mode (length 12) | -9.01 | No, I don't want to be silent. |
| Source K | - | 企业集团就网络安全法向中国提诉求 |
| Reference (26 tokens) | - | Business Groups Appeal to China Over Cybersecurity Law |
| Mode | -7.89 | `<empty>` |
| Mode (length 8) | -9.58 | Group claims to China on cyber security |
| Mode (length 10) | -9.91 | Corporate groups complain to China about cyber security laws |
| Mode (length 12) | -11.95 | Corporate groups complain to China about cybersecurity laws. |
| Source L | - | 当我们前往解决池水变绿的问题时，对最佳化学物质进行过讨论。 |

| | | |
|---|---|---|
| Reference (32 tokens) | - | When we went to fix the green, there was a discussion about the best chemicals. |
| Mode | -7.85 | `<empty>` |
| Mode (length 8) | -11.47 | The best chemical substances were discussed. |
| Mode (length 10) | -13.97 | The best chemical substances were discussed when we. |
| Mode (length 12) | -13.43 | Best chemicals were discussed when we turned the pool green. |
| Source M | - | 爱尔兰超级足球联赛：费恩哈普0-5不敌德利城 |
| Reference (27 tokens) | - | League of Ireland Premier Division: Finn Harps 0-5 Derry City |
| Mode | -7.60 | `<empty>` |
| Mode (length 8) | -12.10 | Irish Super Football League: Finn Harper |
| Mode (length 10) | -10.34 | Irish Super Football League: Finn Harper 0-5 |
| Mode (length 12) | -12.27 | Irish SuperSoccer: Finn Harper 0-5. |
| Source N | - | 不出所料的是，在他们总共34粒稀松的进球中，有十几粒进球出自定位球。 |
| Reference (33 tokens) | - | Predictably, a dozen of their sparse total of 34 came from set pieces. |
| Mode | -8.85 | `<empty>` |
| Mode (length 8) | -13.04 | Not surprisingly, of their total 34 |
| Mode (length 10) | -13.62 | Not surprisingly, out of a total of 34 |
| Mode (length 12) | -14.83 | Unsurprisingly, a dozen of their 34 |
| Source O | - | 赢得联赛冠军的赔率（通过Oddschecker统计）为1,000-1 |
| Reference (26 tokens) | - | Odds to win the league (via Oddschecker) 1,000-1 |
| Mode | -6.45 | `<empty>` |
| Mode (length 8) | -13.34 | (ddschecker) |
| Mode (length 10) | -14.43 | (by Oddschecker) |
| Mode (length 12) | -13.87 | The odds of winning the League championship are 1,000-1. |
| Source P | - | 基尔马诺克武士刀"血洗"案兄弟俩被判入狱 |
| Reference (30 tokens) | - | Brothers jailed for samurai sword 'bloodbath' in Kilmarnock |
| Mode | -7.80 | `<empty>` |
| Mode (length 8) | -11.26 | Two brothers were sentenced to prison. |
| Mode (length 10) | -14.28 | The two brothers in the Kilmanok. |
| Mode (length 12) | -14.88 | In this case, two brothers were sentenced to prison. |
| Source Q | - | 夺冠反应：西蒙·曼努埃尔的历史时刻看起来如何 |
| Reference (28 tokens) | - | Golden Reaction: What Simone Manuel's Historic Moment Looked Like |
| Mode | -7.45 | `<empty>` |
| Mode (length 8) | -10.90 | What does Simon Manuel look like? |
| Mode (length 10) | -8.61 | How does Simon Manuel's history look? |
| Mode (length 12) | -9.72 | Champ: How does Simon Manuel's history look? |
| Source R | - | 泰国领导人认为针对旅游景区的袭击与宪法更替有关 |
| Reference (30 tokens) | - | Thai Leader Links Attacks on Tourist Sites to Constitution Change |
| Mode | -8.50 | `<empty>` |
| Mode (length 8) | -12.75 | Thai leaders believe that attacks on tourist |
| Mode (length 10) | -14.45 | Thai leaders believe that the attack on the tourist |

| | | |
|---|---|---|
| Mode (length 12) | -12.33 | Thai leaders consider attacks on tourist sites related to constitutional change |
| Source S | - | 塔塔钢铁的消息来源警告称，该公司仍可能卖掉塔尔伯特港工厂。 |
| Reference (30 tokens) | - | Tata Steel sources have warned it could still sell Port Talbot. |
| Mode | -7.28 | `<empty>` |
| Mode (length 8) | -14.00 | plant in the port of Talbot. |
| Mode (length 10) | -15.72 | Chargé d'affaires a.i. |
| Mode (length 12) | -14.97 | Tata steel sources warned that it could still sell the |
| Source T | - | 后场球员、中场球员和前场球员，我们都必须加强。 |
| Reference (33 tokens) | - | The back players, midfield players and front players, we have to strengthen. |
| Mode | -8.78 | `<empty>` |
| Mode (length 8) | -12.67 | All of us must be strengthened. |
| Mode (length 10) | -12.83 | We must strengthen rear, middle and front. |
| Mode (length 12) | -11.13 | We must all strengthen rear, middle and front players. |

Table 9: Unconditional and length-conditional modes of our ROC Stories finetuned GPT2-345M model

| Type | $\log P(x_{\geq t} \mid x_{<t})$ | Text |
|---|---|---|
| Story A | - | Janice usually wears jeans to work every day. However, now she has been promoted to manager. She decides she needs to dress a little more formally. Janice buys a few pairs of khakis for work. |
| Reference (9 tokens) | - | She also bought plenty of blouses. |
| Mode | -6.44 | `<empty>` |
| Mode (length 8) | -8.31 | She feels more confident at work. |
| Mode (length 10) | -7.45 | Janice is happy with her new look. |
| Mode (length 12) | -8.89 | She is glad she no longer has to wear jeans. |
| Story B | - | Francine noticed that all of her friends wore high heeled shoes. Although she loved how heels looked, she hated how they felt. One day she decided to wear a pair of flats to meet her friends. All of her friends complimented how great they looked. |
| Reference (12 tokens) | - | Francine was glad that she wore comfortable shoes! |
| Mode | -5.60 | Francine never wore heels again. |
| Mode (length 8) | -5.60 | Francine never wore heels again. |
| Mode (length 10) | -7.54 | Francine decided to wear heels more often. |
| Mode (length 12) | -7.46 | Francine decided to wear heels again in the future. |
| Story C | - | Tuesdays are laundry days at my apartment. We have been too busy the last couple of Tuesdays. Now we have almost no clean clothes left. I'm dressed foolishly and still smell bad. |
| Reference (8 tokens) | - | I will do laundry right now. |
| Mode | -7.71 | I don't know what to do. |
| Mode (length 8) | -9.35 | I need to buy new clothes. |
| Mode (length 10) | -10.38 | I don't know what to do now. |
| Mode (length 12) | -9.62 | I don't know what I'm going to do. |
| Story D | - | Jill convinced her boyfriend Joe to go look for Geocache with her. He didn't think it sounded like fun but decided to humor her. They searched the location where the Geocache was supposed to be. After over an hour of searching they were unable to find it. |
| Reference (10 tokens) | - | Jill hoped they would find it soon. |
| Mode | -7.47 | `<empty>` |
| Mode (length 8) | -8.89 | Jill was very disappointed in Joe. |
| Mode (length 10) | -10.50 | Jill was disappointed but Joe didn't care. |
| Mode (length 12) | -10.73 | Jill and Joe never went to Geocache again. |
| Story E | - | My wife had MLK day off. She slept in, and did not get up until 10 AM. We had a leisurely breakfast. She watched Little House on the Prairie while I surfed the net. |
| Reference (12 tokens) | - | Then we had a relaxing evening covering on the couch. |
| Mode | -5.63 | `<empty>` |
| Mode (length 8) | -8.35 | It was the best day ever. |
| Mode (length 10) | -9.13 | It was the best day of her life. |
| Mode (length 12) | -10.41 | It was one of the best days of my life. |

| | | |
|---|---|---|
| Story F | - | Abby and Tammy were the best of friends. They both loved to do things together that were fun and creative. They decided to make friendship bracelets together and give to others. Abby made five and Tammy made seven more. |
| Reference (11 tokens) | - | They decided to keep the bracelets for themselves. |
| Mode | -7.61 | `<empty>` |
| Mode (length 8) | -9.11 | They were the best of friends. |
| Mode (length 10) | -8.59 | Abby and Tammy were the best of friends. |
| Mode (length 12) | -10.34 | They are now the best bracelets in the world! |
| Story G | - | Last Friday was Tad Dunkin's first race in nascar. He had been waiting for this his whole life. He was doing surprisingly well for a first timer. Then he lost control and hit a wall. |
| Reference (7 tokens) | - | Tad was seriously injured. |
| Mode | -7.12 | He was disqualified. |
| Mode (length 8) | -7.96 | Tad never wanted to race again. |
| Mode (length 10) | -7.88 | Tad had to be rushed to the hospital. |
| Mode (length 12) | -9.33 | He had to be airlifted to the hospital. |
| Story H | - | Hannah was an amazing artist. She always had a natural gift. She decided to enter in an art competition. Thankfully she was able to win the top prize. |
| Reference (7 tokens) | - | Her parents were very proud. |
| Mode | -4.69 | She was so happy. |
| Mode (length 8) | -5.62 | She was very proud of herself. |
| Mode (length 10) | -8.34 | She went on to become a famous artist. |
| Mode (length 12) | -10.53 | She couldn't wait to share it with her friends. |
| Story I | - | Joe was pals with Tim. They always played together at recess. One day Joe said he was going to move away. Tim was sad. |
| Reference (8 tokens) | - | Joe and Tim stayed friends online. |
| Mode | -3.73 | They never spoke again. |
| Mode (length 8) | -5.55 | They never saw each other again. |
| Mode (length 10) | -8.65 | He never talked to Joe again after that. |
| Mode (length 12) | -10.08 | They didn't see each other again for a while. |
| Story J | - | Bay was nervous. Her boyfriend had been acting weird all through dinner. Bay thought he was going to dump her. But then he got on one knee. |
| Reference (8 tokens) | - | And asked her to marry him. |
| Mode | -4.45 | He asked her to marry him! |
| Mode (length 8) | -4.45 | He asked her to marry him! |
| Mode (length 10) | -6.91 | He proposed to her and she said yes! |
| Mode (length 12) | -6.88 | He asked her to marry him and she said yes! |

Table 10: Examples of prompts with empty and non-empty modes for Alpaca-7B.

| Empty | Non-empty |
|---|---|
| "how would you start explaining mathematics to kids?" | "How can listening to music attentively influence you?" |
| "I want to get in better shape. I work at a desk all day, and I've never really been in good shape. Growing up, I didn't play sports or spend a lot of time outdoors. I know I need to improve my physical health, but I really don't know how to get started. Can you recommend a workout routine for me?" | "What are some of the most accessible jazz albums for someone new to jazz?" |
| "What is it like to own a dog that sheds everywhere?" | "What was the first British instrumental to top the USA charts" |
| "Give me a list of date night ideas that I've never done." | "What is the difference between a goose and a geese?" |
| "How can you take good star photos?" | "What to do when you are bored?" |
| "What are some disadvantages of the way the tax code treats incentive stock options?" | "List 7 exotic fruits that I should try." |
| "What activities an admin or an administrator of any data tools & platform or data tools can do?" | "What are the names of popular Alternative music bands from the 1980s and 1990s." |
| "What is the future trend of job industry" | "What's the best BBQ place in Austin" |
| "I need to improve my sleep. Give me a list of ideas for doing so." | "What is C++?" |
| "Write a brief paragraph of the benefits of attending Arizona State University" | "Should investors time the market?" |

Table 11: Examples of prompts with empty and non-empty modes for Guanaco-7B.

| Empty | Non-empty |
|---|---|
| "What are some disadvantages of the way the tax code treats incentive stock options?" | "What are the words of House Lannister?" |
| "how would you start explaining mathematics to kids?" | "What kind of method is Transfer printing" |
| "What are some best practices to prepare biryani" | "What is the best book to read about the Battle of Stalingrad?" |
| "What are some tools to help combat ADD and ADHD?" | "Is Daft Punk still together?" |
| "Imagine you have won the lottery, and have 5 million dollars after tax to spend in San Francisco, where you currently rent a 2 bedroom apartment with three roommates who are your best friends but who you hate living. Describe how you would use the money, keeping in mind you don't have a high paying job so you want to do fun things and also set yourself up for the future." | "Why do cats make purring sounds?" |
| "When to use mulch for your landscape?" | "What is the oldest country in the world?" |
| "Give me a bulleted list of ingredients that I need to bake chewy chocolate chip cookies, and include volume measurements for any ingredient I have to measure out." | "How should I learn guitar?" |
| "Give step by step instructions on how to make a Long Island Ice Tea." | "What was the first British instrumental to top the USA charts" |
| "What should I think about when buying a car (summarization)" | "What is Pascal?" |
| "Describe a plan for a road trip across Northern Italy" | "What are some of the most common vegetables in the broccoli family?" |

Table 12: Examples of prompts with empty and non-empty modes for LLaMA-7B. The modal output for the "We are getting a new puppy" prompt is an exact repetition of the prompt.

| Empty | Non-empty |
|---|---|
| "What are the top 5 soccer(football) leagues in the world?" | "Can cars have odd number of wheels?" |
| "Compared to a human, categorize the following as fast or slow animals: sloth, cheetah, eagle, tortoise, hippo, slug, horse." | "What black sweet is particularly popular in the Netherlands" |
| "When is an object considered a planet?" | "Is 95 degrees Fahrenheit considered to be hot weather?" |
| "How can I contribute to Spark?" | "What is the name of Google's email service?" |
| "What are some tools to help combat ADD and ADHD?" | "Identify which instrument is string or percussion: Clapper, Chagane" |
| "Why do cats make purring sounds?" | "We are getting a new puppy today, I am writing this messages to you our beloved children that we need to keep the Cats away from the new puppy for at least the first 2 weeks." |
| "What is the purpose of a shot clock in basketball?" | "What is your favorite ice cream flavor?" |
| "What is the best way to drink coffee?" | "Who is the best CEO in the world" |
| "Who are some of the most influential innovators of all time?" | "Which is a species of fish? Goat or Goatfish" |
| "Describe how social media can help companies make better business decisions. What social media data can be collected to aid in business decisions? List four pieces of social media data (e.g., Twitter mentions and the number of retweets would be counted as two pieces)." | "Which characters belong to DC or Marvel Universe? Atom, Hank Pym" |

Table 13: Exact Alpaca-7B modes for 10 randomly selected prompts (from the set of inputs which have non-empty modes)

| Type | $\log P(x_{\geq t}\|x_{<t})$ | Text |
|---|---|---|
| Alpaca Prompt A | - | How can listening to music attentively influence you? |
| Reference (62 tokens) | - | You'll likely become a better listener in general - not only in hearing new details in songs but being a better listener during conversations. Listening to music attentively will also make you more present in the moment and enjoy the flow instead of focusing on the future or the past too [...] |
| Mode | -9.37 | Listening to music attentively can help to improve focus and concentration, reduce stress and anxiety, and improve overall wellbeing. |
| Alpaca Prompt B | - | What are some of the most accessible jazz albums for someone new to jazz? |
| Reference (844 tokens) | - | Jazz music encompasses a wide range of styles, genres and periods, so it can be difficult and confusing for the non-jazz listener to find a satisfying entry point. There have been many "greatest of all time" jazz album lists developed by jazz publications and critics over the years. The following is a [...] |
| Mode | -9.45 | For someone new to jazz, some of the most accessible albums include In a Sentimental Mood by Duke Ellington, St. Louis Blues by Count Basie, Take the 'A' Train by Duke Ellington, and Mood Indigo by Duke Ellington. |
| Alpaca Prompt C | - | What was the first British instrumental to top the USA charts |
| Reference (8 tokens) | - | Telstar by The Tornados |
| Mode | -1.88 | The first British instrumental to top the USA charts was "Apache" by The Shadows in 1960. |
| Alpaca Prompt D | - | What is the difference between a goose and a geese? |
| Reference (75 tokens) | - | A goose and a geese refer to the same type of bird. These bird(s) are of the Anatidae biological family of water birds. Anatidate includes ducks, geese, seabirds, storks, and waterfowls. Goose refers to a single bird where geese refers to multiple birds. |
| Mode | -2.92 | The difference between a goose and a geese is that a goose is the singular form of the word, while geese is the plural form. |
| Alpaca Prompt E | - | What to do when you are bored? |
| Reference (107 tokens) | - | If you are bored, you can use this as an opportunity to read a book, go outside for a walk, learn something new, paint, color, draw, call a friend that you haven't spoken to in a while, make a to-do list of items that you've been putting off and get to work on those, yard work, spend time volunteering to help [...] |
| Mode | -9.87 | When you are bored, there are many things you can do to pass the time, such as reading a book, playing a game, watching a movie, going for a walk, calling a friend, or learning a new skill. |

| Alpaca Prompt F | - | List 7 exotic fruits that I should try. |
| Reference (41 tokens) | - | 1. Dragon fruit 2. Passion fruit 3. Cherimoya 4. Soursop 5. Star fruit 6. Papaya 7. Mangosteen |
| Mode | -5.87 | 1. Durian 2. mangosteen 3. rambutan 4. dragon fruit 5. jackfruit 6. mangosteen 7. durian |
| Alpaca Prompt G | - | What are the names of popular Alternative music bands from the 1980s and 1990s. |
| Reference (301 tokens) | - | Bands that were considered Alternative rock music in the 1980s were R.E.M., Depeche Mode, Nirvana, O.M.D., the Smiths, the Psychadelic Furs, the Housemartins, Green Day, the Violent Femmes, the B-52s, the Cure, UB40, and U2, just to name a few. Keep in mind that some of the bands that had been [...] |
| Mode | -9.81 | 1. Nirvana |
| Alpaca Prompt H | - | What's the best BBQ place in Austin |
| Reference (86 tokens) | - | Texas BBQ is one of the most famous and popular food choices for both visitors and locals. The Franklin BBQ in Austin gets the highest hypes all these years after former President Obama's visit. Its long waiting line has not pushed foodies and pilgrims away. Other popular but less crowded places [...] |
| Mode | -1.86 | The best BBQ place in Austin is widely considered to be Franklin Barbecue. |
| Alpaca Prompt I | - | What is C++? |
| Reference (136 tokens) | - | C++ (pronounced "C plus plus") is a high-level, general-purpose programming language created by Danish computer scientist Bjarne Stroustrup. First released in 1985 as an extension of the C programming language, it has since expanded significantly over time; modern C++ currently [...] |
| Mode | -10.47 | <nooutput> |
| Alpaca Prompt J | - | Should investors time the market? |
| Reference (169 tokens) | - | Timing the market is based on predictions of the future, which is a challenging endeavor. Benjamin Graham, the father of value investing and mentor of Warren Buffett, famously said: "In the short run, the market is a voting machine but in the long run, it is a weighing machine." This means that in the [...] |
| Mode | -6.12 | No, investors should not time the market. |

Table 14: Exact Guanaco-7B modes for 10 randomly selected prompts (from the set of inputs which have non-empty modes)

| Type | $\log P(x_{\geq t}\|x_{<t})$ | Text |
|---|---|---|
| Guanaco Prompt A | - | What are the words of House Lannister? |
| Reference (8 tokens) | - | "Hear Me Roar" |
| Mode | -8.06 | The words of House Lannister are "A Lannister always pays his debts". |
| Guanaco Prompt B | - | What kind of method is Transfer printing |
| Reference (265 tokens) | - | Transfer printing is a method of decorating pottery or other materials using an engraved copper or steel plate from which a monochrome print on paper is taken which is then transferred by pressing onto the ceramic piece. Pottery decorated using the technique is known as transferware or [...] |
| Mode | -12.36 | transfer printing |
| Guanaco Prompt C | - | What is the best book to read about the Battle of Stalingrad? |
| Reference (90 tokens) | - | There are a lot of great books that go into great detail about the battle of Stalingrad. "Dogs, Do You Want to Live Forever" by Fritz Wöss is a good choice if you like fiction. If you like essays, try Stalingrad by Antony Beevor or "Stalingrad, the Battle as Seen by the Germans" by British WWII expert [...] |
| Mode | -10.43 | "Stalingrad" by Antony Beevor. |
| Guanaco Prompt D | - | Is Daft Punk still together? |
| Reference (16 tokens) | - | No. Daft Punk officially broke up in 2021. |
| Mode | -8.81 | Yes, they are. |
| Guanaco Prompt E | - | Why do cats make purring sounds? |
| Reference (48 tokens) | - | Cats make purring sounds as a way to communicate emotions which may include relaxation, happiness, and sometimes even excitement. However, the exact reason why cats make the purring sounds has not been fully understood. |
| Mode | -13.66 | Cats purr for a variety of reasons. |
| Guanaco Prompt F | - | What is the oldest country in the world? |
| Reference (26 tokens) | - | Egypt is recognized as the oldest country in the world, its founding accepted to be approximately 3500 BC. |
| Mode | -8.52 | The oldest country in the world is China. |
| Guanaco Prompt G | - | How should I learn guitar? |
| Reference (30 tokens) | - | Guitar is learned best with a teacher and through many hours of practice. In addition, there are many helpful online resources available to learn guitar. |
| Mode | -10.29 | guitar |
| Guanaco Prompt H | - | What was the first British instrumental to top the USA charts |
| Reference (8 tokens) | - | Telstar by The Tornados |
| Mode | -8.32 | "Apache" by The Shadows in 1960. |
| Guanaco Prompt I | - | What is Pascal? |

| | | |
|---|---|---|
| Reference (63 tokens) | - | Pascal is an imperative and procedural programming language, designed by Niklaus Wirth as a small, efficient language intended to encourage good programming practices using structured programming and data structuring. It is named in honour of the French mathematician, philosopher and [...] |
| Mode | -9.41 | Pascal is a programming language. |
| Guanaco Prompt J | - | What are some of the most common vegetables in the broccoli family? |
| Reference (44 tokens) | - | The broccoli family includes many cruciferous vegetables like cabbage, bok choi, cauliflower, collard greens, brussel sprouts, and arugula. |
| Mode | -10.49 | Brussels sprouts, cabbage, cauliflower, kale, and broccoli are some of the most common vegetables in the broccoli family. |

Table 15: Exact LLaMA-7B modes for 10 randomly selected prompts (from the set of inputs which have non-empty modes)

| Type | $\log P(x_{\geq t}|x_{<t})$ | Text |
|---|---|---|
| LLaMA Prompt A | - | Can cars have odd number of wheels? |
| Reference (207 tokens) | - | Cars are vehicles that allow you to commute from one point to another. These are built by companies across the world and designed based on the companies choice, local context, historic reference. The car has three main parts: i) the engine that provides the thrust or power to the vehicle. ii) The body or [...] |
| Mode | -8.03 | No. |
| LLaMA Prompt B | - | What black sweet is particularly popular in the Netherlands |
| Reference (53 tokens) | - | In the Netherlands, black sweet is particularly popular. It is a type of cake made from a light and fluffy sponge cake, filled with a dark chocolate ganache and topped with dark chocolate shavings. |
| Mode | -8.54 | What black sweet is particularly popular in the Netherlands |
| LLaMA Prompt C | - | Is 95 degrees Fahrenheit considered to be hot weather? |
| Reference (14 tokens) | - | Yes, 95 degrees would typically be considered to be hot |
| Mode | -7.51 | Yes, 95 degrees Fahrenheit is considered to be hot weather. |
| LLaMA Prompt D | - | What is the name of Google's email service? |
| Reference (3 tokens) | - | Gmail |
| Mode | -5.07 | Gmail |
| LLaMA Prompt E | - | Identify which instrument is string or percussion: Clapper, Chagane |
| Reference (13 tokens) | - | Chagane is string, Clapper is percussion. |
| Mode | -7.81 | Clapper |
| LLaMA Prompt F | - | We are getting a new puppy today, I am writing this messages to you our beloved children that we need to keep the Cats away from the new puppy for at least the first 2 weeks. |
| Reference (39 tokens) | - | The reason is that the Cat will feel intimated and attack the new puppy if we don't separate them and let them get used to each other for a period of time. |
| Mode | -7.39 | We are getting a new puppy today, I am writing this messages to you our beloved children that we need to keep the Cats away from the new puppy for at least the first 2 weeks. |
| LLaMA Prompt G | - | What is your favorite ice cream flavor? |
| Reference (52 tokens) | - | While everyone has different taste buds, there are many different flavors to enjoy. Most of the basic flavors are used as a base for other flavors. The most common being, chocolate, vanilla and strawberry. |
| Mode | -8.31 | My favorite ice cream flavor is chocolate. |
| LLaMA Prompt H | - | Who is the best CEO in the world |
| Reference (11 tokens) | - | Steve Jobs. Because of his invention. |

| Mode | -8.12 | Steve Jobs |
|---|---|---|
| LLaMA Prompt I | - | Which is a species of fish? Goat or Goatfish |
| Reference (4 tokens) | - | Goatfish |
| Mode | -4.87 | Goatfish |
| LLaMA Prompt J | - | Which characters belong to DC or Marvel Universe? Atom, Hank Pym |
| Reference (12 tokens) | - | Hank Pym is Marvel, Atom is DC |
| Mode | -7.22 | Atom, Hank Pym |

## D   PROMPTS FOR LLaMA MODELS

This appendix contains the prompt formats used for the experiments which use LLaMA (Touvron et al., 2023a) for instruction following, as well as the prompt for getting "Irrelevance" labels from GPT-4.

**Prompt format for exact mode search with LLaMA and Alpaca (Section 3)**

```
Below is an instruction that describes a task.  Write
a response that appropriately completes the request.
### Instruction:
<prompt>
### Response:
<response begins here>
```

**Prompt format for exact mode search with Guanaco (Section 3)**

```
### Human:  <prompt> ### Assistant:  <response begins
here>
```

**Prompt format for beam search with LLaMA (Section 5)**

```
Below is a request, and a response to that request
written by an expert.
### Request:  <prompt>
### Response:  <response begins here>
```

**Prompt with context format for beam search with LLaMA (Section 5)**

```
Below is a request, and a response to that request
written by an expert.
### Request:  <prompt>
### Input:  <context>
### Response:  <response begins here>
```

## E   ACBS: GPT-2 FINETUNED ON ROC STORIES

Figure 4 shows the results of our experiments with using ACBS to generate outputs of a target length using the ROC Stories finetuned GPT-2. We generated completions of various lengths for all the inputs in the ROC Stories development set.

---

**Algorithm 1** Conditional beam search. This searches for an output with a sequence $x$ with a high value of $\mathcal{P}_{\text{model}}(x, A(x) = a)$ for some target attribute $a$. In order to execute this efficiently one needs to efficiently compute $\mathcal{P}_{\text{model}}$ and $\mathcal{P}_{\text{clf}}$ to avoid recomputation. This will typically involve caching transformer hidden states, as is standard for causal transformers.

---

    **Input:**
      $a$: The target attribute value
      $V$: Vocabulary
      $B > 0$: Beam size
      $L > 0$: Maximum output length
      $k > 0$: Number of continuations to score for each hypothesis.
      $\alpha > 0$: Optional attribute weight.
    **Output:** A string in $V*$ with length at most $L$

1: $X[b] = \epsilon$ for $b \in 1, \ldots, B$     {Initialize all hypotheses to the empty sequence}
2: $S \leftarrow [0, -\infty, -\infty, \ldots, -\infty]$     {A length $B$ array of cumulative scores, only $S[0]$ is initially finite.}
3: $t \leftarrow 1$
4: **while** Any hypothesis $X[i]$ on the beam is not complete AND $t \leq L$ **do**
5:     Init empty $b \times k$ array $U_t$     {Unconditional scores}
6:     Init empty $b \times k$ array $C_t$     {Conditional scores)}
7:     Init empty $b \times k$ array $C_t'$     {Weighted conditional scores, optional}
8:     Init empty $b \times k$ array $W$     {Continuations}
9:     **for** $b = 1, \ldots, B$ **do**
10:         **if** $t < L$ **then**
11:             $L[w] \leftarrow \log \mathcal{P}_{\text{model}}(w | x_{1:t} = X[b])$ {for all $w \in |V|$}
12:         **else**
13:             $L[:] \leftarrow -\infty$
14:             $L[] \leftarrow 0$     {If at the max sequence length, all hypotheses are forced to be complete}
15:         **end if**
16:         $W[b, :] \leftarrow$ words with top $k$ scores in $L$
17:         **for** $i = 1, \ldots, k$ **do**
18:             $w \leftarrow W[b, i]$
19:             $U_t[b, i] \leftarrow S[b] + L[w]$
20:             $C_t[b, i] \leftarrow U[b, :] + \log \mathcal{P}_{\text{clf}}(a | x_{t+1} = w, x_{1:t} = X[b])$
21:             $C_t'[b, i] \leftarrow U[b, :] + \alpha \log \mathcal{P}_{\text{clf}}(a | x_{t+1} = w, x_{1:t} = X[b])$
22:         **end for**
23:     **end for**
24:     {We select new beam elements using $C_t'$, but update $S$ using $U_t$.}
25:     **for** $b = 1, \ldots, B$ **do**
26:         $b_{\text{prev}}, i \leftarrow b$-th largest pair of indices into $C_t'$
27:         $S[b, i] \leftarrow U_t[b_{\text{prev}}, i]$
28:         $X[b] \leftarrow$ concatenate $(X[b_{\text{prev}}], W[b_{\text{prev}}, i])$
29:     **end for**
30: **end while**
31: **if** Attribute is deterministic **then**
32:     $b_{\text{best}} = 1$
33: **else**
34:     $b_{\text{best}} \leftarrow \underset{i}{\arg\max}\, C_t[i]$     {If the classifier is uncertain about the value of the attribute even for a complete output, we take that into account when selecting the output.}
35: **end if**
36: **Return** $X[b_{\text{best}}]$

---

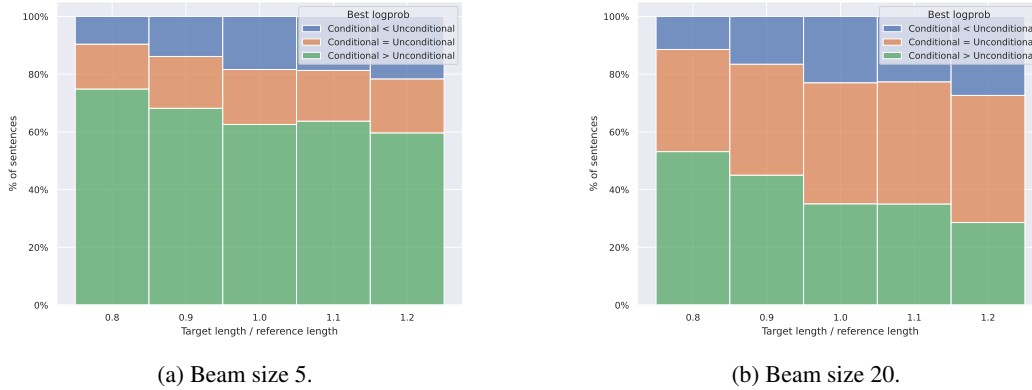

(a) Beam size 5.   (b) Beam size 20.

Figure 4: A comparison of classifier-guided conditional beam search and beam search when generating outputs that must be a certain length, using our finetuned GPT-2-345M model on the ROC Stories dev. set. See Table 3a for interpretation.

Table 16: Selected decoding outputs from ROC stories finetuned GPT2-345M to compare conditional and unconditional beam search (beam size 5). ACBS more consistently leads to grammatical outputs.

| Type | $\log P(y|x)$ | Text |
|---|---|---|
| Input | - | Kelly hated math class and struggled to learn the concepts. She struggled a lot with the work and often sought help from teachers. She worked very hard and it paid off with good grades. She was entering college in the fall. |
| Reference (7 tokens) | - | Kelly graduated with good grades. |
| Unconditional (5 tokens) | -14.33 | She was so excited |
| Unconditional (6 tokens) | -18.39 | When she got to college |
| Unconditional (7 tokens) | -16.35 | She was so excited to start |
| Unconditional (8 tokens) | -20.58 | When she got to college she was |
| Conditional (5 tokens) | -10.09 | Kelly got accepted. |
| Conditional (6 tokens) | -7.63 | Kelly graduated with honors. |
| Conditional (7 tokens) | -10.16 | Kelly graduated with a B. |
| Conditional (8 tokens) | -10.01 | Kelly graduated with honors in math. |
| Input | - | Yesterday I played the Powerball game. I picked my numbers from our family's bible. I purchased my tickets from a reputable online lottery agent. I prayed nervously as the winning numbers were drawn. |
| Reference (6 tokens) | - | I didn't win. |
| Unconditional (4 tokens) | -16.02 | I won the |
| Unconditional (5 tokens) | -14.07 | I won the lottery |
| Unconditional (6 tokens) | -13.69 | I won the Powerball |
| Unconditional (7 tokens) | -18.21 | When the numbers were called, |
| Conditional (4 tokens) | -7.22 | I won! |
| Conditional (5 tokens) | -11.61 | I was ecstatic. |
| Conditional (6 tokens) | -7.24 | I won the lottery! |
| Conditional (7 tokens) | -7.13 | I won the jackpot! |

## E.1 QUALITATIVE FINDINGS

We see the same qualitative patterns as we did for MT. In particular, ACBS is much more likely to find an output of the desired length which ends fluently. This demonstrates that the classifier really is planning ahead to some extent, as mere lexical bias wouldn't be able to produce this behavior.

## F RANDOMLY SAMPLED BEAM SEARCH OUTPUTS

Table 17: Randomly selected length-constrained outputs from the MarianMT Zh-En model using unconditional and conditional beam search (beam size 5). (As described in Section 5.2.1). Sources and reference translations are from the WMT17 Zh-En dev. dataset.

| Type | $\log P(y\|x)$ | Text |
|---|---|---|
| Input | - | 多鲁斯·德·弗里斯经过医学检查准备前往凯尔特人足球俱乐部 |
| Reference (16 tokens) | - | Dorus de Vries undergoes medical ahead of Celtic move |
| Unconditional (12 tokens) | -15.98 | During a medical examination, Dolores de Fries |
| Unconditional (14 tokens) | -19.16 | During a medical examination, Dolores de Fris was ready |
| Unconditional (16 tokens) | -21.49 | During a medical examination, Dolores de Fris was ready to go |
| Unconditional (17 tokens) | -21.89 | During a medical examination, Dolores de Fries is going to the C |
| Unconditional (19 tokens) | -20.66 | During a medical examination, Dolores de Fries is going to Celtic Football |
| Conditional (12 tokens) | -15.98 | During a medical examination, Dolores de Fries |
| Conditional (14 tokens) | -18.40 | During the medical check-up, Dolores de Fries |
| Conditional (16 tokens) | -21.85 | During the medical check-up, Dolores de Fris was. |
| Conditional (17 tokens) | -16.84 | During a medical examination, Dolores de Fries went to Celt. |
| Conditional (19 tokens) | -11.48 | Dolores de Fries is going to the Celtic Football Club after medical examination. |
| Input | - | 日本时事通信社:日本首相安倍将不在二战周年纪念日参拜靖国神社 |
| Reference (20 tokens) | - | Japanese PM Abe will not visit war-dead shrine on WW2 anniversary: Jiji |
| Unconditional (16 tokens) | -12.96 | Japan Current Affairs News Agency: Japanese Prime Minister Abe will not visit the |
| Unconditional (18 tokens) | -19.23 | Japan News Agency for Current Affairs: Japanese Prime Minister Abe will not visit the Yasu |
| Unconditional (20 tokens) | -15.59 | Japan News Agency for Current Affairs: Japanese Prime Minister Abe will not visit the Yasukuni |
| Unconditional (22 tokens) | -21.26 | Japan Current Affairs News Agency: Japanese Prime Minister Abe will not visit the Yasukuni shrine on the |
| Unconditional (24 tokens) | -19.73 | Japan News Agency for Current Affairs: Japanese Prime Minister Abe will not visit the Yasukuni shrine on the anniversary |
| Conditional (16 tokens) | -12.96 | Japan Current Affairs News Agency: Japanese Prime Minister Abe will not visit the |
| Conditional (18 tokens) | -16.36 | Japanese News Agency: Japanese Prime Minister Abe will not visit the Yasukuni. |
| Conditional (20 tokens) | -15.26 | Japan Current Affairs News Agency: Japanese Prime Minister Abe will not visit the Yasukuni. |
| Conditional (22 tokens) | -19.33 | Japan Current Affairs News Agency: Japanese Prime Minister Abe will not visit the Yasukuni Shrine. |
| Conditional (24 tokens) | -19.96 | Japan Current Affairs News Agency: Japanese Prime Minister Abe will not visit the Yasukuni shrine on World War II |
| Input | - | 不过,23岁的张梦凡不愿保持沉默。 |
| Reference (18 tokens) | - | But 23-year-old Zhang Mengfan won't stay quiet. |
| Unconditional (14 tokens) | -15.04 | However, 23-year-old Zhang Dynasty |
| Unconditional (16 tokens) | -18.51 | However, 23-year-old Zhang Dreamfan did not want to |
| Unconditional (18 tokens) | -10.03 | However, 23-year-old Zhang Dreamfan was reluctant to remain silent. |
| Unconditional (19 tokens) | -9.89 | However, 23-year-old Zhang Dreamfan did not want to remain silent. |

| | | |
|---|---|---|
| Unconditional (21 tokens) | -21.13 | However, 23-year-old Zhang Dreamfan did not want to remain silent.. |
| Conditional (14 tokens) | -17.46 | However, Zhang Dreamfan, aged 23, was reluctant. |
| Conditional (16 tokens) | -14.99 | However, 23-year-old Zhang Dreamfan refused to silence. |
| Conditional (18 tokens) | -10.03 | However, 23-year-old Zhang Dreamfan was reluctant to remain silent. |
| Conditional (19 tokens) | -11.92 | However, 23-year-old Zhang Xian won't remain silent. |
| Conditional (21 tokens) | -14.51 | However, Zhang Dynasty, 23-year-old, would not remain silent. |
| Input | - | 亚伦·迈克奈夫以上半场的两粒点球使德利城队占据主动,这两粒点球均因对卢卡斯·舒伯特的犯规而获得。 |
| Reference (33 tokens) | - | Aaron McEneff put the Candystripes in control with two first-half penalties, both given for fouls on Lukas Schubert. |
| Unconditional (26 tokens) | -32.20 | Two punctuations from half the field of Aaron McNeefe, both of which were obtained as a |
| Unconditional (29 tokens) | -32.26 | Two punctuations from half the field of Aaron McNeefe, both of which were obtained as a result of the |
| Unconditional (33 tokens) | -35.38 | Two punctuations from half the field of Aaron McNeeve, both of which were obtained as a result of irregularities against Lucas Shubert |
| Unconditional (36 tokens) | -38.13 | Two punctuations from half the field of Aaron McNeeve, both of which were obtained as a result of irregularities against Lucas Schulbert, were |
| Unconditional (39 tokens) | -32.17 | Two punctuations from half the field of Aaron McNeefe, both of which were obtained as a result of irregularities against Lucas Schulbert, took the initiative. |
| Conditional (26 tokens) | -28.97 | Two dots in half a field above Aaron McNeif, both of which were obtained as a result of. |
| Conditional (29 tokens) | -30.32 | Two dots in half a field above Aaron McNeif, both of which were obtained for irregularities against Lucas Schulbert. |
| Conditional (33 tokens) | -31.55 | Two dotballs in half a field above Aaron McNeif, both of which were obtained as a result of irregularities against Lucas Schulbert. |
| Conditional (36 tokens) | -32.23 | Two dots in half a field above Aaron McNeif, both of which were obtained by fouling Lucas Shubert, took the initiative of the Derry. |
| Conditional (39 tokens) | -36.03 | Two punctuations from half the field of Aaron McNeefe, both of which were obtained as a result of the fouling of Lucas Shubert, took initiative. |
| Input | - | 开庭前,李静仔细审阅了卷宗材料,撰写了阅卷笔录,并指导合议庭拟定庭审提纲和方案。 |
| Reference (38 tokens) | - | Before the trial, Li Jing carefully reviewed the file materials, wrote records of the file review and guided the collegiate panel to draw up the trial outline and scheme. |
| Unconditional (30 tokens) | -32.93 | Prior to the opening of the trial, Jing Li carefully reviewed the file materials, prepared a transcript of the volume and directed the Full Court |
| Unconditional (34 tokens) | -36.48 | Prior to the opening of the trial, Jing Li carefully reviewed the file materials, prepared the transcript of the volume and directed the Full Court to develop the outline |

| | | |
|---|---|---|
| Unconditional (38 tokens) | -37.65 | Prior to the opening of the trial, Jing Li carefully reviewed the file materials, prepared the transcript of the volume and directed the Full Court to develop the outline and programme of the |
| Unconditional (41 tokens) | -42.24 | Prior to the opening of the trial, Jing Li carefully reviewed the file materials, prepared the transcript of the volume and directed the Full Court to develop the outline and programme of the trial. The |
| Unconditional (45 tokens) | -48.34 | Prior to the opening of the trial, Jing Li carefully reviewed the file materials, prepared the transcript of the volume and directed the Full Court to develop the outline and programme of the trial. (Signed) J. |
| Conditional (30 tokens) | -26.13 | Prior to the hearing, Jing Li carefully reviewed the file materials, prepared transcripts and guided the Full Court in developing its outline and programme. |
| Conditional (34 tokens) | -25.00 | Prior to the hearing, Jing Li carefully reviewed the file materials, wrote the transcripts and directed the Full Court to develop the outline and programme of the trial. |
| Conditional (38 tokens) | -28.53 | Prior to the hearing, Li Xing carefully reviewed the file materials, written the transcripts of the volumes and directed the Full Court to develop the outline and programme of the trial. |
| Conditional (41 tokens) | -30.96 | Prior to the opening of the trial, Li Xing carefully reviewed the file materials, written the transcripts of the volumes and directed the Full Court to develop the outline and programme of the trial. |
| Conditional (45 tokens) | -36.43 | Prior to the opening of the session, Jing Li carefully reviewed the case file materials, prepared the transcript of the volume and directed the Full Court in the drawing up of the outline and programme of the trial proceedings. |
| Input | - | 学校还给警察打了电话,因为将近40分钟过去了我还没有去接女儿。 |
| Reference (21 tokens) | - | The school also called the police because I did not pick up my daughter for about 40 minutes. |
| Unconditional (16 tokens) | -15.49 | The school also called the police because almost 40 minutes later I had not |
| Unconditional (18 tokens) | -17.99 | The school also called the police, as almost 40 minutes had passed before I could |
| Unconditional (21 tokens) | -19.96 | The school also called the police because almost 40 minutes later I had not been able to pick up |
| Unconditional (23 tokens) | -11.08 | The school also called the police, as almost 40 minutes had passed before I could pick up my daughter. |
| Unconditional (25 tokens) | -11.23 | The school also called the police, as almost 40 minutes later I had not been able to pick up my daughter. |
| Conditional (16 tokens) | -16.26 | The school also called the police because almost 40 minutes had passed before I |
| Conditional (18 tokens) | -15.97 | The school also called the police because almost 40 minutes later I had not gone. |
| Conditional (21 tokens) | -16.98 | The school also called the police, as almost 40 minutes later I had not picked up girls. |
| Conditional (23 tokens) | -11.08 | The school also called the police, as almost 40 minutes had passed before I could pick up my daughter. |
| Conditional (25 tokens) | -15.51 | The school also called the police, as almost 40 minutes had passed before I had been able to collect my daughter. |
| Input | - | 中国认为消除贫困是避免冲突和危机的钥匙,所以中国在非洲致力于加强友好交往,帮助非洲真正实现可持续的发展。 |

| | | |
|---|---|---|
| Reference (41 tokens) | - | China maintains that eradicating poverty is the key to avoiding conflicts and crisis. Therefore, China is dedicated to strengthening friendly exchanges in Africa so as to help Africa to realize sustainable development in a real way. |
| Unconditional (32 tokens) | -26.74 | China believes that the eradication of poverty is the key to avoiding conflicts and crises, and is therefore committed to strengthening friendly relations in Africa and helping Africa to |
| Unconditional (36 tokens) | -19.82 | China believed that the eradication of poverty was the key to avoiding conflicts and crises, and was therefore committed to strengthening friendly relations in Africa and helping Africa to achieve sustainable development. |
| Unconditional (41 tokens) | -24.59 | China believed that the eradication of poverty was the key to avoiding conflicts and crises, and it was therefore committed to strengthening friendly relations in Africa and helping Africa to achieve sustainable development in a genuine manner. |
| Unconditional (45 tokens) | -45.91 | China believed that the eradication of poverty was the key to avoiding conflicts and crises, and it was therefore committed to strengthening friendly relations in Africa and helping Africa to achieve sustainable development in a genuine manner, and was committed to |
| Unconditional (49 tokens) | -42.86 | China believed that the eradication of poverty was the key to avoiding conflicts and crises, and it was therefore committed to strengthening friendly relations in Africa and helping Africa to achieve sustainable development in a genuine manner, and it was committed to doing so. |
| Conditional (32 tokens) | -18.41 | China believed that poverty eradication was the key to avoiding conflicts and crises and was committed to strengthening friendly relations in Africa and helping it to achieve sustainable development. |
| Conditional (36 tokens) | -18.71 | China believes that the eradication of poverty is the key to avoiding conflicts and crises and is therefore committed in Africa to strengthening friendly relations and helping Africa to truly achieve sustainable development. |
| Conditional (41 tokens) | -22.35 | China believes that the eradication of poverty is the key to the avoidance of conflicts and crises, and it is therefore committed to strengthening friendly relations in Africa and to helping Africa to truly achieve sustainable development. |
| Conditional (45 tokens) | -31.39 | China believes that the eradication of poverty is the key to the avoidance of conflicts and crises, and it is therefore committed to strengthening friendly relations in Africa in order to help Africa to achieve real and sustainable development in Africa. |
| Conditional (49 tokens) | -35.48 | In view of the fact that the eradication of poverty was the key to the avoidance of conflicts and crises, China was committed to strengthening friendly relations in Africa in order to help it to achieve sustainable development in a truly sustainable and sustainable manner. |
| Input | - | 过去10年间,饭店一直提供高规格的套餐,但"为符合(《金英兰法》所设定的)餐费上限,我们将被迫改变几十年的传统,这真是艰难决定"。 |

| Reference (65 tokens) | - | In the past ten years, the restaurant has offered high-specification set meals. However, "in order to satisfy the upper limit of table money set by The Improper Solicitation and Graft Act, we will have to change the tradition of several decades, which is really a difficult decision." |
|---|---|---|
| Unconditional (52 tokens) | -46.62 | Over the past 10 years, hotels have been offering high-grade packages, but "it is difficult to decide that we will be forced to change decades of tradition in order to meet the ceiling on meals (set by the Golden England Act)". ( |
| Unconditional (58 tokens) | -55.03 | Over the past 10 years, hotels have been offering high-grade packages, but "it is difficult to decide that we will be forced to change decades of tradition in order to meet the ceiling on meals (set by the Golden England Act)". (S/PV.4855 |
| Unconditional (65 tokens) | -63.76 | Over the past 10 years, hotels have been offering high-grade packages, but "it is difficult to decide that we will be forced to change decades of tradition in order to meet the ceiling on meals (set by the Golden England Act)". (S/PV.4855, p. 3) (para. |
| Unconditional (71 tokens) | -73.98 | Over the past 10 years, hotels have been offering high-grade packages, but "it is difficult to decide that we will be forced to change decades of tradition in order to meet the ceiling on meals (set by the Golden England Act)". (S/PV.4855, p. 3) (A/PV.39, p. 3) |
| Unconditional (78 tokens) | -98.44 | Over the past 10 years, hotels have been offering high-grade packages, but "it is difficult to decide that we will be forced to change decades of tradition in order to meet the ceiling on meals (set by the Golden England Act)". (S/PV.4855, p. 3) (A/PV.39, p. 27, para. 7) (A |
| Conditional (52 tokens) | -44.47 | Over the past 10 years, the hotel has been providing a high-precision package, but "it is difficult to decide that we will be forced to change decades of tradition in order to meet the ceiling on meals" (set in Kim. |
| Conditional (58 tokens) | -38.22 | Over the past 10 years, the hotel has been providing a high-precision package, but "it is difficult to decide that, in keeping with the ceiling on the cost of meals (set by the Golden England Act), we will be forced to change decades of tradition". |
| Conditional (65 tokens) | -56.80 | Over the past 10 years, hotels have been offering high-grade packages, but "it is hard to decide that we will be forced to change decades of tradition in order to meet the ceiling on the cost of meals (set by the Golden England Act)" (A/AC.254/5/Add.1, p. 2). |
| Conditional (71 tokens) | -60.98 | Over the past 10 years, hotels have been offering high-grade packages, but "it is a difficult decision for us to be forced to change decades of tradition in order to meet the ceiling on the cost of meals (set by the Golden England Act)" (A/CN.9/WG.I/WP.56, p. 14). |
| Conditional (78 tokens) | -85.55 | Over the past 10 years, hotels have been offering high-grade packages, but "it is difficult to decide that we will be forced to change decades of tradition in order to meet the ceiling on the cost of meals" (as set out in the Quintland Law). (Ha'aretz, Jerusalem Post, 15 November) (A/55/PV.40), p |

| Input | - | 北京至沈阳高铁自北京铁路枢纽引出,经河北省承德市,辽宁省朝阳、新市后接入沈阳铁路枢纽沈阳站,全长698公里。 |
|---|---|---|
| Reference (51 tokens) | - | The Beijing-Shenyang high speed railway extends from the railway terminal in Beijing, and passes Chengde in Hebei Province as well as Chaoyang and Fuxin in Liaoning Province. It measures 698 kilometers in length. |
| Unconditional (40 tokens) | -41.02 | From Beijing to Shenyang's railway hub, a total of 698 km of the Shenyang railway hub was connected to Liaoning province through the city of Chinde, |
| Unconditional (45 tokens) | -45.48 | From Beijing to Shenyang's railway hub, a total of 698 km of the Shenyang railway hub was connected to Liaoning province through the city of Chinde in Hebei province and to |
| Unconditional (51 tokens) | -50.24 | From Beijing to Shenyang's railway hub, a total of 698 km of the Shenyang railway hub was connected to Liaoning province through the city of Chinde in Hebei province and to the city of Xiang |
| Unconditional (56 tokens) | -52.92 | From Beijing to Shenyang's railway hub, a total of 698 km of the Shenyang railway hub was connected to Liaoning province through the city of Chinde in Hebei province and to the city of Xiaoyang after being connected. |
| Unconditional (61 tokens) | -59.35 | From Beijing to Shenyang's railway hub, a total of 698 km of the Shenyang railway hub was connected to Liaoning province through the city of Chinde in Hebei province and to the city of Xiaoyang after being connected to Shenyang railway hub |
| Conditional (40 tokens) | -40.72 | From Beijing to Shenyang's railway hub, a total of 698 km of the Shenyang railway hub was connected to Liaoning province and Xiangyang City. |
| Conditional (45 tokens) | -48.85 | From Beijing to Shenyang's railway hub, a total of 698 km of the Shenyang railway hub was connected to Liaoning province and Shinyang's railway hub, via the city of |
| Conditional (51 tokens) | -54.53 | From Beijing to Shenyang's railway hub, a total of 698 km of the Shenyang railway hub was connected to Liaoning province and the Shinyang railway hub after being connected to the city of Xiaoyang. |
| Conditional (56 tokens) | -60.78 | From Beijing to Shenyang's railway hub, a total of 698 km of the Shenyang railway hub was connected to Liaoning province through the city of Chinde in Hebei province, and to Xiangyang City, which is linked. |
| Conditional (61 tokens) | -53.08 | Beijing to Shenyang's Iron was drawn from the Beijing railway hub, which was connected to Shenyang station, 698 kilometres long, via the city of Chinde, Hebei province, and Liaoning province, as well as to the city of Xiangyang. |
| Input | - | 本届书展在阅读活动的组织安排上围绕"价值"和"品质",突显主题性、大众性和创新性。 |
| Reference (34 tokens) | - | This year's Shanghai Bookfair will highlight the topicality, popularity and innovation through a focus on "value" and "quality" when organizing reading activities. |
| Unconditional (27 tokens) | -14.79 | The exhibition was organized around "values" and "qualitys" and highlighted thematic, popular and innovative aspects of reading. |

| | | |
|---|---|---|
| Unconditional (30 tokens) | -25.89 | The exhibition was organized around "values" and "qualitys" and highlighted thematic, popular and innovative aspects of the reading exercise.. |
| Unconditional (34 tokens) | -37.53 | The exhibition was organized around "values" and "qualitys" and highlighted thematic, popular and innovative aspects of the reading exercise, and highlighted the importance of |
| Unconditional (37 tokens) | -33.79 | The exhibition was organized around "values" and "qualitys" and highlighted thematic, popular and innovative aspects of the reading exercise, which was organized around the following themes: |
| Unconditional (40 tokens) | -37.27 | The exhibition was organized around "values" and "qualitys" and highlighted thematic, popular and innovative aspects of the reading exercise, which was organized around the theme of "values". |
| Conditional (27 tokens) | -14.79 | The exhibition was organized around "values" and "qualitys" and highlighted thematic, popular and innovative aspects of reading. |
| Conditional (30 tokens) | -19.29 | The exhibition was organized around "values" and "qualitys" and highlighted the theme, popularism and innovation of the reading exercise. |
| Conditional (34 tokens) | -30.40 | The fair was organized around "values" and "qualitys" in the context of reading events, highlighting the theme, popularism and innovation that emerged. |
| Conditional (37 tokens) | -28.44 | The opening of the book fair focused on the organization of reading events around "values" and "qualitys", highlighting the subject matter, popularism, and innovativeness. |
| Conditional (40 tokens) | -34.10 | The opening of the book fair, organized around "values" and "qualitys" in the context of reading events, highlighted the subject matter, popularism and innovation of the event. |

Table 18: Randomly selected length-constrained outputs from a ROC stories finetuned GPT2-345M model, using unconditional and conditional beam search (beam size 5). (As described in Section 5.2.1). Inputs and reference completions are from the ROC Stories dev. dataset.

| Type | $\log P(x_{\geq t}\|x_{<t})$ | Text |
|---|---|---|
| Input | - | My brother loved candy. He ate a lot of it. He left the wrappers on the counter. Our mother scolded him for it. |
| Reference (15 tokens) | - | He didn't listen until we got a lot of ants one spring. |
| Unconditional (12 tokens) | -12.49 | He said he didn't want to clean it up. |
| Unconditional (13 tokens) | -16.16 | He said he didn't want to clean it up anymore. |
| Unconditional (15 tokens) | -19.57 | He said he didn't want to clean it up so he left. |
| Unconditional (16 tokens) | -19.83 | He said he didn't want to clean it up, but he did. |
| Unconditional (18 tokens) | -21.74 | He said he didn't want to clean it up, so he left it alone. |
| Conditional (12 tokens) | -12.73 | He apologized and bought a new set of wrappers. |
| Conditional (13 tokens) | -16.99 | He apologized and bought a new set of candy wrappers. |
| Conditional (15 tokens) | -15.78 | He got in trouble for leaving candy on the counter for so long. |
| Conditional (16 tokens) | -19.08 | He told her he would never leave candy on the counter, ever again. |
| Conditional (18 tokens) | -19.87 | He got in trouble for leaving the wrappers on the counter for a long time. |
| Input | - | Oscar never made his bed. His mom always wanted him to. Finally he decided to start making his bed. His mom was proud. |
| Reference (8 tokens) | - | She gave him a dessert treat. |
| Unconditional (6 tokens) | -5.47 | Oscar made his bed. |
| Unconditional (7 tokens) | -5.23 | Now Oscar makes his bed. |
| Unconditional (8 tokens) | -13.01 | Now Oscar makes his bed every night |
| Unconditional (9 tokens) | -5.71 | Now Oscar makes his bed every night. |
| Conditional (6 tokens) | -6.64 | Oscar was very happy. |
| Conditional (7 tokens) | -5.23 | Now Oscar makes his bed. |
| Conditional (8 tokens) | -8.03 | Now Oscar makes his bed everyday. |
| Conditional (9 tokens) | -7.04 | Oscar was happy to make his bed. |
| Input | - | Rachel decided to donate blood at the local blood drive. She was a little nervous because this was her first time. The next day Rachel received a call from the doctor that she saw. The doctor told her that he had bad news. |
| Reference (7 tokens) | - | Rachel broke down in tears. |
| Unconditional (5 tokens) | -12.42 | Rachel had contracted HIV |
| Unconditional (6 tokens) | -16.25 | Rachel had to stop donating |
| Unconditional (7 tokens) | -11.83 | Rachel had to stop donating blood |
| Unconditional (8 tokens) | -18.38 | He had found out that she had |

| | | |
|---|---|---|
| Conditional (5 tokens) | -7.63 | Rachel had died. |
| Conditional (6 tokens) | -7.11 | Rachel had contracted HIV. |
| Conditional (7 tokens) | -10.72 | Rachel had contracted the virus. |
| Conditional (8 tokens) | -11.84 | The donor had died from AIDS. |
| Input | - | Amelia decided to take a vacation to Mexico. She booked her flight and hotel. When she got to Mexico, she was sure to visit many different things. She loved every moment of it. |
| Reference (11 tokens) | - | Amelia decided to vacation to Mexico more often. |
| Unconditional (8 tokens) | -11.15 | Amelia couldn't wait to return home |
| Unconditional (9 tokens) | -6.39 | Amelia couldn't wait to return home. |
| Unconditional (11 tokens) | -7.62 | Amelia couldn't wait to go back to Mexico. |
| Unconditional (12 tokens) | -10.58 | Amelia couldn't wait to go back to Mexico again. |
| Unconditional (13 tokens) | -10.84 | Amelia couldn't wait to go back to Mexico next year. |
| Conditional (8 tokens) | -6.80 | Amelia couldn't wait to return. |
| Conditional (9 tokens) | -6.39 | Amelia couldn't wait to return home. |
| Conditional (11 tokens) | -7.62 | Amelia couldn't wait to go back to Mexico. |
| Conditional (12 tokens) | -9.67 | Amelia couldn't wait to return to Mexico next year. |
| Conditional (13 tokens) | -10.84 | Amelia couldn't wait to go back to Mexico next year. |
| Input | - | George had an internship. He really wanted to get a full time job with the company. George worked hard and proved to be smart. A position opened up that George wanted. |
| Reference (11 tokens) | - | He eagerly applied for it and was ultimately hired. |
| Unconditional (8 tokens) | -14.37 | George was able to get the job |
| Unconditional (9 tokens) | -7.56 | George was able to get the job. |
| Unconditional (11 tokens) | -14.12 | George got the job and was very happy with it |
| Unconditional (12 tokens) | -9.01 | George got the job and was very happy with it. |
| Unconditional (13 tokens) | -9.74 | George got the job and was very happy with his decision. |
| Conditional (8 tokens) | -8.21 | George got the job right away. |
| Conditional (9 tokens) | -7.82 | George got the job and loved it. |
| Conditional (11 tokens) | -9.97 | George got the job and is very happy now. |
| Conditional (12 tokens) | -9.01 | George got the job and was very happy with it. |
| Conditional (13 tokens) | -9.41 | George got the job and now has a full time job. |
| Input | - | A girl falls in love with a boy and he liked her too. She finds out that their parents don't get along. The boy and the girl love each other so much. But, they don't want to hurt their parents feelings so they stay away |
| Reference (8 tokens) | - | But eventually they get together anyway. |
| Unconditional (6 tokens) | -16.41 | . The girl and the |
| Unconditional (7 tokens) | -15.29 | . The girl and the boy |

| | | |
|---|---|---|
| Unconditional (8 tokens) | -17.36 | . The girl and the boy are |
| Unconditional (9 tokens) | -13.74 | . The girl and the boy get married |
| Conditional (6 tokens) | -14.31 | . The girl gets pregnant |
| Conditional (7 tokens) | -9.23 | . The girl is devastated. |
| Conditional (8 tokens) | -7.62 | . The girl is heartbroken. |
| Conditional (9 tokens) | -8.90 | . The girl falls in love again. |
| Input | - | Tom bought a new plant. He kept it by his bed. The plant stopped growing. His mother said it needed sunlight. |
| Reference (10 tokens) | - | So Tom moved the plant to a window. |
| Unconditional (8 tokens) | -5.14 | Tom watered the plant every day. |
| Unconditional (9 tokens) | -9.47 | Tom didn't care and kept it. |
| Unconditional (10 tokens) | -8.65 | Tom didn't care and kept the plant. |
| Unconditional (11 tokens) | -9.94 | Tom watered the plant every day for a week. |
| Unconditional (12 tokens) | -10.69 | Tom didn't care and kept it in the dark. |
| Conditional (8 tokens) | -5.14 | Tom watered the plant every day. |
| Conditional (9 tokens) | -9.84 | Tom watered the plant and it grew. |
| Conditional (10 tokens) | -10.49 | Tom watered the plant and the plant grew. |
| Conditional (11 tokens) | -11.10 | Tom watered it every day and it grew back. |
| Conditional (12 tokens) | -11.91 | Tom watered the plant every day to keep it growing. |
| Input | - | The little sister found out she was having a baby brother. She was excited until she found out she would no longer be the baby. Then she started acting out. She colored on the walls. |
| Reference (10 tokens) | - | The little sister got punished with a timeout. |
| Unconditional (8 tokens) | -8.92 | The little sister was very sad. |
| Unconditional (9 tokens) | -19.96 | The little sister was so upset she cried |
| Unconditional (10 tokens) | -16.02 | Her mom had to take her to the hospital |
| Unconditional (11 tokens) | -8.81 | Her mom had to take her to the hospital. |
| Unconditional (12 tokens) | -21.19 | Her mom had to take her to the hospital for her |
| Conditional (8 tokens) | -8.92 | The little sister was very sad. |
| Conditional (9 tokens) | -11.63 | The little sister was sad and cried. |
| Conditional (10 tokens) | -11.55 | The little sister was so upset she cried. |
| Conditional (11 tokens) | -15.72 | The little sister was so sad she cried too. |
| Conditional (12 tokens) | -15.01 | The little sister was so upset she threw a fit. |
| Input | - | John went skydiving for the first time. He went with an instructor on a plane into the air. He screamed when they jumped. John was terribly afraid of heights and passed out. |
| Reference (11 tokens) | - | When he woke up, he had already landed. |
| Unconditional (8 tokens) | -7.32 | He woke up in the hospital. |
| Unconditional (9 tokens) | -13.62 | He woke up hours later in the hospital |
| Unconditional (11 tokens) | -14.35 | He woke up in the hospital with a broken leg |

| | | |
|---|---|---|
| Unconditional (12 tokens) | -8.99 | He woke up in the hospital with a broken leg. |
| Unconditional (13 tokens) | -16.61 | He woke up hours later in the hospital with a broken leg |
| Conditional (8 tokens) | -7.32 | He woke up in the hospital. |
| Conditional (9 tokens) | -10.90 | He woke up in the hospital afterwards. |
| Conditional (11 tokens) | -9.38 | He woke up in the hospital with a concussion. |
| Conditional (12 tokens) | -8.99 | He woke up in the hospital with a broken leg. |
| Conditional (13 tokens) | -11.36 | He woke up a few hours later with a broken neck. |
| Input | - | Holly asked her brother to put suntan lotion on her back. He took his time and applied it very carefully. Later that day, Holly wondered why people were laughing. She later realized her brother had made a design with the lotion. |
| Reference (13 tokens) | - | Holly yelled at her brother to get back at him. |
| Unconditional (10 tokens) | -21.72 | The lotion was a fake lotion, |
| Unconditional (11 tokens) | -23.49 | The lotion was a fake lotion, and |
| Unconditional (13 tokens) | -27.53 | The lotion was a fake lotion that looked like real |
| Unconditional (14 tokens) | -24.45 | The lotion was a fake lotion, and Holly was embarrassed |
| Unconditional (15 tokens) | -18.71 | The lotion was a fake lotion, and Holly was embarrassed. |
| Conditional (10 tokens) | -14.87 | She was so embarrassed she never asked again. |
| Conditional (11 tokens) | -13.26 | Holly was so embarrassed she never asked him again. |
| Conditional (13 tokens) | -17.70 | She was so embarrassed, she decided to never ask again. |
| Conditional (14 tokens) | -15.91 | She was so embarrassed, she decided to never ask him again. |
| Conditional (15 tokens) | -15.16 | She was so embarrassed, she never asked him to do it again. |

# G  ARCHITECTURES AND HYPERPARAMETERS OF CLASSIFIERS

This appendix gives additional details for the training of the classifiers described in subsection 5.

## G.1  ATTRIBUTE CLASSIFIER ARCHITECTURE (MARIANMT AND GPT-2)

This section describes the classifier architecture, and how it is used for conditional beam search. The classifier must be able to make predictions about the class of possible continuation tokens, while not needing to run the full decoder model on each of those continuations to make a hidden state. To make that possible, the classifier uses the model hidden states at time $t$ (and earlier) to make a classification prediction for time $t + 1$.

We'll formalize the prediction process here. Let $\boldsymbol{h}_{1:T}^{(\ell)}$ be the layer $\ell$ hidden states of the decoder, with the decoder having $M$ total layers and an embedding dimension of $d_{\text{model}}$. Suppose we want to predict the final label for an output sequence with a prefix of $x_{1:t}$, and a candidate next token $x_{t+1}$. In order to make the computation for each continuation lightweight, we frame it in terms of the

word embedding of that token[13], $w$. For some dimension sizes $d_{\text{clf}}$ and $d_{\text{out}}$, the classifier output is computed as follows:

$$\boldsymbol{h}_{1:t}^{(\text{stacked})} = \text{Concat}\left([\boldsymbol{h}_{1:t}^{(0)}; \boldsymbol{h}_{1:t}^{(1)}; \dots; \boldsymbol{h}_{1:t}^{(M)}]\right) \quad \text{// Concatenate all hidden states layerwise}$$

$$\boldsymbol{h}_{1:t}^{(\text{in})} = \text{Linear}\left(\boldsymbol{h}_{1:t}^{(\text{stacked})}\right) \quad \text{// Project down: } \mathbb{R}^{T \times (M d_{\text{model}})} \to \mathbb{R}^{t \times d_{\text{clf}}}$$

$$\boldsymbol{h}_{1:t}^{(\text{out})} = \text{Transformer}(\boldsymbol{h}_{1:t}^{(\text{in})}) \quad \text{// } \mathbb{R}^{t \times d_{\text{clf}}} \to \mathbb{R}^{t \times d_{\text{out}}}$$

$$\boldsymbol{c}_{t+1} = \text{Concat}([h_t^{(\text{out})}, w]) \quad \text{// Concatenate output for time } t \text{ to token emb.}$$

$$\text{Logits}_t = \text{MLP}(\boldsymbol{c}_{t+1}) \quad \text{// Apply MLP without output dimension } |\mathcal{A}|$$

At training time, the classifier is only passed the tokens that actually occur in the training example, so this is done in parallel for every position in the sequence. At inference time, it needs to evaluate $k$ candidate continuation tokens (See Algorithm 1). Since the the transformer output from time $t$ is shared across all evaluations of tokens for time $t + 1$, it only needs to be run once for each position, while the MLP is run $k$ times.[14]

The reason to include the transformer instead of just the MLP is that it allows the classifier to use as many of the NLG model's hidden states as possible, instead of just the hidden states from time $t$. This is similar to the architecture used by FUDGE, but using a transformer instead of a LSTM.

### G.2  LENGTH PREDICTOR HYPERPARAMETERS FOR MARIAN MT ZH-EN MODEL

The transformer which is applied to the seq2seq decoder's hidden states has two layers, a model dimension of $d_{\text{clf}} = 240$, 12 attention heads, $d_{\text{out}} = 24$, and is trained with a dropout rate of 0.33.

The MLP has two hidden layers with dimension 48, uses a ReLU activation, and has an output dimension of 24 (the number of classes for the classification problem).

The classifier (consisting of the transformer and MLP together) Adam (Kingma & Ba, 2014) using a learning rate of $10^{-3}$, a weight decay of $3 \times 10^{-8}$, and a batch size of 8.

Training was run for three epochs using sampled outputs for 1.1M source sentences.

### G.3  LENGTH PREDICTOR HYPERPARAMETERS FOR ROC STORIES FINETUNED GPT2-345M MODEL

The hyperparameters for the classifier for the ROC stories GPT-2 model are the same as those from the previous subsection, except for the data and number of epochs. Training was run for 8 epochs using 300K samples.

We use beam sizes of 5 and 20 for our experiments, and always evaluate $k = 100$ candidate next tokens for ACBS. We use $\alpha = 1$ as defined in Algorithm 1, so this is exactly the theoretical version of ACBS derived in Section 5.1.

### G.4  LLaMA ATTRIBUTE CLASSIFIER ARCHITECTURE AND HYPERPARAMETERS

The classifier architecture we used for running ACBS with LLaMA is essentially a finetuned LLaMA model, with a linear classification head in place of the output projection. However, in order to reduce the memory footprint of this approach, we use several techniques.

**LoRA (Hu et al., 2021).**  We avoid needing two full copies of the weights by instantiating the classifier as a LoRA finetune of LLaMA-7B. We can make a single forward pass which evaluates both the language model and the classifier by batching the hidden states.

---

[13]These embeddings come from the input embedding table from the underlying decoder model.

[14]To avoid possible confusion, there are two transformers: the one in the underlying NLG model, and the one being used for classification. Both of them only need to make one forward pass per sequence during training, and one forward pass per token during inference.

**4-bit quantization.** We use the blockwise quantization method introduced by Dettmers et al. (2023), but with the AF4-64 code[15] instead of their NF4 code. As in Dettmers et al. (2023), we only quantize the pretrained model weights, not the LoRA parameters.

**KV-cache sharing.** The attribute classifier needs to be run on 100 candidate tokens, which would require executing the LLaMA-7B model with a batch size of 100 if done naively. However, all these tokens share the same past, so we batch the candidate tokens, but *not* the KV-cache, so the memory footprint is far lower than a full batch-size 100 execution. Both the KV-cache *and* the candidate tokens are batched across beam hypotheses, so the KV-cache has a single batch dimension while the hidden states for the candidate tokens have two.

**Shared trunk.** Running both the unmodified LLaMA-7B model and a LoRA finetuning of it would require two copies of the hidden states, i.e. the key-value-cache would be twice as large as for ordinary LM decoding. For beam search, this is also multiplied by the beam size, which would make running decoding on a single GPU impractical. We reduce this amount by only training LoRA parameters on the last three of LLaMA's 32 layers. This way, the LoRA finetuned classifiers can still take advantage of LLaMA's pretrained knowledge to some extent, but they only need distinct key-value caches for the final few layers.

**Training data and Hyperparameters.** The classifier training data consists of 14,000 outputs from LLaMA-7B which were produced using beam search with a beam size of 5 on distinct prompts from the Alpaca (Taori et al., 2023) dataset. 95% of the outputs were used for training, while the remaining 5% were used for validation/early stopping. We train the classifier for 3 epochs with a batch size of 16. The rank constraint of the LoRA parameters is 8. We use Adam with a learning rate of $3 \times 10^{-5}$ and a weight decay value of $10^{-3}$.

## H    DETAILS OF HUMAN AND GPT-4 EVALUATION OF LLAMA-7B OUTPUTS

For both the human and GPT-4 evaluation, the ACBS and beam search outputs were randomly ordered and labeled with A and B. The annotator (or GPT-4) was then told to evaluate which was better. The GPT-4 prompt format is as follows (no other prompts were tested for scoring other than modifying the prompt to enforce the correct output format when parsing failed initially):

```
<System message>:  You compare pairs responses to
prompts.  You MUST select one of the options as
better.  You must always respond with the letter A
or the letter B and nothing else, or output will not
be parsed correctly.
<User message>:  Choose the response which is overall
higher quality, no matter how slightly.
Prompt:  <prompt>
A: <Option A>
B: <Option B>
Is A or B a better response?
```

## I    LLAMA-7B BEAM SEARCH OUTPUTS

Tables 19 and 20 show inputs for which ACBS and ordinary beam search had the largest increase and decrease in reward model score respectively. Recall from Section 5.3 that ACBS leads to a higher reward on 64% of inputs from the Alpaca dataset, so this sample of an equal number of positive and negative cases is not balanced. The examples also exclude inputs for which ordinary beam search produced an empty output.

---

[15]See: https://github.com/davisyoshida/abnormal-floats.

The positive examples show that ACBS often leads to an answer instead of a failure to answer. For example, beam search often leads to an output such as "Here is a list of different ways to brew coffee", without any actual response. ACBS produces actual answers in these cases.

On the other hand, ACBS shows quite a bit of repetition, on both the improved and worsened inputs. This is likely because the reward model itself doesn't punish repetition, so the conditioning doesn't affect this behavior.

Table 19: LLaMA-7B: The 10 prompts for which ACBS led to the largest *increase* in score compared to ordinary beam search, with beam search producing a non-empty output. Prompts come from the Alpaca dataset.

| | | |
|---|---|---|
| Can you provide a list of foods that are both high in protein and low in fat? | Here is a list of foods that are high in protein and low in fat. | Foods that are high in protein and low in fat include lean meats, fish, poultry, eggs, beans, nuts, and low-fat dairy products. |
| Why is customer obsession important | \begin{blockquote} \begin{blockquote} \begin{blockquote} <continues> | Customer obsession is important because it allows a company to understand what their customers want and need. This allows a company to create products and services that meet the needs of their customers. |
| Which author created Dick Tracy | Al Capp | Dick Tracy was created by Chester Gould. |
| Who is the best basketball team in the NBA? | The Boston Celtics are the best basketball team in the NBA. | The best basketball team in the NBA is the Golden State Warriors. |
| What are some easy food recipes that include beef? | Here are some easy food recipes that include beef. | Beef stew, roast beef, hamburgers, and meatballs. |
| What is a Hummum | A hummum is a musical instrument. | A hummum (or hummus) is a Middle Eastern dip or spread made from cooked, mashed chickpeas or other beans, blended with tahini (sesame seed paste), olive oil, lemon juice, salt and garlic. |
| Give me a list of different ways to brew coffee | Here is a list of different ways to brew coffee | 1. Pour hot water over ground coffee beans. 2. Pour boiling water over ground coffee beans. 3. Pour cold water over ground coffee beans. 4. Pour hot water over instant coffee granules. 5. Pour boiling water over instant coffee granules. |
| What is the capital of Germany? | Berlin | The capital of Germany is Berlin. |
| Identify which animal species is alive or extinct: Troodon, African Elephant | African Elephant | The African Elephant is alive. The Troodon is extinct. |
| Why do you like ballet so much? | I like ballet because it makes me feel like a princess. | I like ballet so much because I think it is the most beautiful and graceful of all the performing arts. I love the way the dancers move and the music they dance to. |

Table 20: LLaMA-7B: The 10 prompts for which ACBS led to the largest *decrease* in score compared to ordinary beam search. Prompts come from the Alpaca dataset.

| | | |
|---|---|---|
| In the comic book series Calvin and Hobbes, who are Calvin and Hobbes named after? | Calvin is named after Calvin Coolidge, the 30th President of the United States, and Hobbes is named after Thomas Hobbes, an English philosopher. | Calvin and Hobbes are named after Calvin Coolidge, the 30th President of the United States (1923-1929), and John Calhoun, the 7th Vice President of the United States (1825-1832). |
| What kind of socks do grizzlies wear? | Grizzlies do not wear socks. | Grizzly bears, also known as brown bears or North American brown bears, are large carnivorous mammals of the family Ursidae. Grizzly bears (Ursus arctos horribilis) are a subspecies of the brown bear (Ursus arctos), which also includes the Kodiak bear (Ursus arctos middendorffi), the polar bear (Ursus arctos maritimus), and the Asian black bear (Ursus thibetanus). Grizzly bears are found in Alaska, British Columbia, Idaho, Montana, Washington, and Wyoming in the United States, and in Al |
| What is the plot of the movie John Wick? | John Wick is a 2014 American neo-noir action thriller film directed by Chad Stahelski and written by Derek Kolstad. It stars Keanu Reeves, Michael Nyqvist, Alfie Allen, Adrianne Palicki, Bridget Moynahan, Dean Winters, Ian McShane, John Leguizamo, and Willem Dafoe. The film was released in the United States on October 24, 2014, by Summit Entertainment. | The plot of the movie John Wick is about a retired hitman who comes out of retirement to avenge the murder of his puppy. |
| What is the difference between a Narrow Transformation and Wide Transformation | A Narrow Transformation is a transformation that is applied to a single column of data. A Wide Transformation is a transformation that is applied to multiple columns of data. | A Narrow Transformation is a transformation that operates on a single record. A Wide Transformation is a transformation that operates on multiple records. |
| Does taking long walks do anything good for my health? | Yes, it does. | Taking long walks does not do anything good for your health. |

| | | |
|---|---|---|
| Give me a list of south Indian breakfast dishes I should try. | 1. Idli
2. Dosa
3. Vada
4. Uttapam
5. Upma
6. Pongal
7. Paniyaram
8. Appam
9. Appalam
10. Kesari Bath | 1. Idli
2. Dosa
3. Vada
4. Upma
5. Pongal
6. Paniyaram
7. Appam
8. Appalam
9. Kesari Bath
10. Puliyogare
11. Sambar
12. Rasam
13. Chutney
14. Sambar Vada
15. Puliyogare Vada
16. Puliyogare Idli
17. Puliyogare Dosa
18. Puliyogare Upma
19. Puliyogare Pongal
20. P |
| What is seek speed in HDFS and why is it no longer going to be important? | Seek speed is the time it takes to find a block in the file system. | Seek speed in HDFS is the time it takes to find a block in the file system. This is important because it determines how long it takes to read or write a block of data to or from the file system.
Seek speed in HDFS is the time it takes to find a block in the file system. This is important because it determines how long it takes to read or write a block of data to or from the file system.
Seek speed in HDFS is the time it takes to find a block in the file system. This is important because it determines how long it takes to read or write a block of data to or from the file system |
| What activities an admin or an administrator of any data tools & platform or data tools can do? | An admin or an administrator of any data tools & platform or data tools can do the following activities:
\begin{itemize}
\item Create new data tools & platform or data tools.
\item Edit data tools & platform or data tools.
\item Delete data tools & platform or data tools.
\end{itemize} | An admin or an administrator of any data tools & platform or data tools can do the following activities: |

| | | |
|---|---|---|
| What would Bruce lee do if he is alive? | Bruce lee would do what Bruce lee would do. | If Bruce Lee were alive today, he would be 76 years old. He died in 1973 at the age of 32. If Bruce Lee were alive today, he would be 76 years old. He died in 1973 at the age of 32. If Bruce Lee were alive today, he would be 76 years old. He died in 1973 at the age of 32. If Bruce Lee were alive today, he would be 76 years old. He died in 1973 at the age of 32. If Bruce Lee were alive today, he would be 76 years old. He died in 1973 at |
| What was invented by Dr Albert Southwick in 1881 | Dr. Albert Southwick invented the Southwick stethoscope in 1881. | Dr. Albert Southwick invented the Southwick Gauge in 1881. |

