# OpenReview forum: "MAP's not dead yet: Uncovering true language model modes by conditioning away degeneracy"
_ICLR.cc/2024/Conference — Submitted to ICLR 2024_

### Official Review · Reviewer_vYmk · 2023-10-28

**Soundness:** 2 fair
**Presentation:** 1 poor
**Contribution:** 2 fair
**Rating:** 3
**Confidence:** 4

**Summary:**

This work proposes ACBS, a modified version of beam search that produces output from the LM by conditioning on external signals, e.g. length. The authors argue that the unexpected behavior of the model is caused by the low-entropy noise sample and derive their proposed method. The experiments are conducted on two tasks (machine translation and story generation) with model scales up to 7B.

**Strengths:**

* The motivation for this work is interesting.
* The authors provide extensive qualitative examples.

**Weaknesses:**

* **Poor Presentation**: The presentation of this work is poorly written. Please proofread your manuscript before submission. Some examples are
  * Section 1: distribution which the mode representsEikema & Aziz (2020) --- the citation should be included in parenthesis and there should be a white space.
  * Section 1: training data data --> training data
  * Footnote 1: the regurgitating the input --> the regurgitating of the input
  * Section 2.1: distributions arbitrarily closely --> distributions arbitrarily close
  * Section 3: translation model Tiedemann & Thottingal (2020) --> translation model (Tiedemann & Thottingal, 2020)
  * Section 3: ROC stories dataset Mostafazadeh et al. (2016) --> ROC stories dataset (Mostafazadeh et al., 2016)
  * Section 3: LLaMA model Touvron et al. (2023) --> LLaMA model (Touvron et al., 2023)
* **Poor Theoretical Analysis**: The theoretical motivations presented in Section 2.1, 2.2, 2.3 are hard to follow. For instance, the authors write "For the SVO translation example", what is SVO? And there should be a proper citation. The mathematical derivations in those sections are not detailed enough, which clearly undermines the quality of this work.
* **Lack of Details in Experiments**: There are necessary details are missing, including how the outputs are obtained from the LM which are then used to train the classifier; number of epochs; number of training samples.
* **Limited Evaluations**: The evaluations are only considered up to 200 tokens which is too short under current literature, e.g. ChatGPT API has 4k length. I strongly suggest the authors to extend their evaluations up to at least 2k length.
* **No human evaluation**: When comparing two decoding methods like in Section 5.2.1, simply using likelihood is not enough. Human evaluations are necessary to include to provide more evidence of the proposed approach.

**Questions:**

N/A

---

> ### Author Response · Authors · 2023-11-21
> **Thank you for your review**
>
> Thank you for your time in reviewing and considering our topic interesting. We address your concerns below and we also have updated the draft to make the presentation cleaner with additional results to support our findings.
>
> ### Poor presentation
> Thank you for the detailed feedback here, we have addressed the problems in the updated draft.
>
> ### Theoretical analysis
> We motivated our problem and analysis via several examples related to various considerations for natural language processing in Section 2. SVO is a common abbreviation used by linguists and NLP researchers to characterize a language that typically arranges information in a “subject verb object” order in the sentences. We have reworded the writing to make it clear we were referring to the preceding example in the text.
>
> Regarding the mathematical derivations not being enough, our goal in Section 2 is simply to show that the bad mode problem can arise in the absence of model error. This does not require an involved calculation, as it’s clear that a large set of valid outputs can easily be dominated by a small set of noise sequences, even with a tiny mixing rate. There is value in presenting this to the community, as it is an underappreciated point. Most prior work on the topic makes the inference that “bad mode” implies “model error”, which has led to a dearth of research on approximate-MAP methods.
>
> ### Human Evaluation
> We have expanded our evaluation to include several other informative metrics for our experiments.
> In section 5.2.1, we primarily focused on likelihood because we wanted to emphasize that constrained beam search is not an appropriate method for performing length-constrained MAP. We found that ACBS finds sequences which are higher likelihood, and the large number of randomly sampled examples we present do show the trend of ACBS finding more grammatical translations. However, we appreciate your concern that his is not sufficiently convincing, so we have added BLEURT/BLEU scores in the updated draft (Table 3b). ACBS outperforms constrained beam search significantly on the BLEURT metric which captures fluency and well-formedness unlike BLEU (which only considers n-gram precision with a heuristic brevity penalty). We also provide perplexity measurements of the translations under Llama2-7B and find that ACBS results in higher quality sequences under this metric as well (Table 3c).
>
> We also provide further evidence that ACBS leads to improved outputs from LLaMA-7B, a model which was not trained for instruction following. Specifically, we perform a blinded pairwise comparison between the outputs under standard beam search and those found by ACBS, using both GPT-4 and one of the authors. In a camera-ready version we will expand the human evaluation.
>
> ### Context length limits for evaluation
> We agree that experiments using more computational resources would be necessary to establish ACBS as a state-of-the-art decoding method. However, we actually view the ACBS experiments as supporting the preceding sections, rather than the other way around. Prior work has suggested that we should abandon MAP-like methods in favor of sampling-based ones, due beam search’s failure to produce good outputs in tasks less constrained than machine translation, and its failure in even MT at large beam sizes. (For example, see Holtzman et al. (2020), Stahlberg and Byrne (2019), and Eikema and Aziz (2020)).
>
> In this work, we use ACBS as a preliminary indication that the choice to drop MAP was premature, as we suggest in the title of the paper. Our ideal outcome from this paper wouldn’t be that everyone stops using nucleus sampling and switches to ACBS tomorrow. Instead, we hope to spur future work on improved MAP-like methods.
>
> ### Lack of details in experiments
> The space constraints made it difficult for us to include extensive details for a lot of aspects of the paper’s contents. In the original draft, we included extensive details in the appendix as well to complement the claims, experiments, and arguments in the main body.
> However, we would like to point out that we attempted to provide sufficient details in the main body of the original draft- sections 5.2, 5.3, 3, and 3.1. Additionally, we provided more extensive details about our experiment setup in the Appendix sections G and D, as well as giving pointers to them in the Reproducibility section.
>
> To the specific question about training examples for the LLaMA-7B classifier, 14,000 training examples were produced using vanilla beam search on inputs from the databricks-dolly-15k dataset.The number of training examples was stated in the body of the text, but we have added it to Appendix G along with the other details to make it easier to find.
>
> Thank you once again for your detailed feedback, we believe it has helped us produce a much better draft, especially through the addition of more evaluations of our system outputs.

---

### Official Review · Reviewer_3nJt · 2023-10-29

**Soundness:** 3 good
**Presentation:** 3 good
**Contribution:** 2 fair
**Rating:** 5
**Confidence:** 3

**Summary:**

This paper shows that degenerate modal outputs are not necessarily an intrinsic property of language models themselves, but rather are likely a result of contamination in the training data. For improving the quality of text decoded from language models, the paper introduce an algorithm called ACBS (attribute-conditional beam search), which adds an additional constraint on the output to avoid the degenerate behavior. The experiment shows that ACBS are better than ordinary beam search and especially contirbutes to ameliorate empty-string degenerate behavior.

**Strengths:**

1. This paper provides a very detailed explanation to the model architecture, algorithms, and experimental data in the appendix, which is very useful for helping readers to understand the paper’s work.
2. Derivation and motivation are clear. Point out the phenomena and causes of degeneracy problem at the beginning and then solve it later.
3. This paper is logical and flowing. The location and analysis of degeneracy problem are given progressively.
4. The analysis of the low-entropy distractor is concise and easy to understand with examples.
5. The experimental setup follows intuition, and argumentation process is basically based on experiments.

**Weaknesses:**

1. Missing experimental data in 5.3. The detailed experiment result about the comparison between ACBS and regular beam search should be compared in a table instead of directly stating the data in the paragraph. Otherwise, your results won't be convincing.
2. In 3.2.1, figure 1a and figure 1b just represents the increase of empty sequence with source length, lack of the curve that shows the decrease of empty output with source length to better support the conclusion.
3. Lack of explaination about the difference between empty mode and empty output.
4. The paper only gives two examples to illustrate that low-entropy distractor outputs and empty outputs have a high log prob in section 2 but does not provide enough mathematical reasoning, so the argument that the degenerate modal behavior is related to the entropy of the set of valid outputs is not strong enough.
5. In section 3, the x-axis and y-axis markings in Figure 1a are not clear and Figure 1a does not offer enough support for the observed phenomenon that the probability of the empty sequence declines as the source length increases. The experiments in section 3 do not indicate the impact of contamination of the training data, which is emphasized in the abstract and conclusion.

**Questions:**

1. At the end of Section 3, how can we conclude that the degenerate modal behavior is related to the entropy of the set of valid outputs?
2. Exact search experiments are meaningful, but the result that exact search performs good is not relevant to Subsequent Chapters. Maybe their relevance should be emphasized by experiments in terms of computational cost.
3. The experiment to prove that ACBS not only benefit from removing empty outputs should be detailed.

---

> ### Author Response · Authors · 2023-11-21
> **Thank you for your review**
>
> Thank you for your time in reviewing this paper and for your kind words about the structure of the paper and argumentation. Below, we address some of the concerns you raised:
>
> ### Presentation and analysis of results
> Thanks for pointing out that the empirical comparison between regular beam search and ACBS doesn’t appear to be detailed from our in-text description in the original draft. In our updated draft, we report the results in Table 5, and additionally.
>
> Please see our overall response to the reviewers for a description of the other newly added evaluations.
>
> ### Elaboration on figures 1a) and 1b) and other terms used in the paper
> We have re-added a figure which was cut to make this explicit. Figure 2 in the appendix now shows that the probability of the empty output declines with source length. Despite this, the rate at which the mode is empty increases, which is not surprising in light of our analysis in Section 2.
>
> The term “empty output” refers to the output of any decoding method being empty, while an “empty mode” describes the case where the empty sequence is the single most likely output.
>
> ### Connection between our results and low-entropy distractors
> The entropy of a distribution is just the average of the negative log likelihood of a sample, so what we do in Section 2.1 is compare how two distributions with different entropies react to a small amount of noise. We did not have space to get into this, but in practice what one needs is high-entropy PLUS the distribution of valid outputs is not too “non-uniform”. Since we don’t have analytic access to the population distribution for realistic datasets, we aren’t trying to show that the bad mode problem only arises from noise in the data. Instead, we just want to give enough argumentation to show that the inference “bad model mode -> the model has mis-estimated the mode of the distribution” is not true. It is important that this point becomes internalized by the community of NLG researchers, as it strongly affects how we interpret prior findings such as Stahlberg and Byrne (2019).
>
> Additionally, we now also refer to Ott et al. (2018) for another example of a low entropy distractor. In their work, adding a small fraction of training examples where the reference translation is a copy of the source sentence leads to an outsized impact on search outputs. This is just what our analysis predicts, but they do make the theoretical argument for why this _must_ happen in certain cases, depending on the noise rate and entropy/likelihood of the valid outputs.
>
> ### Question 1
> > At the end of Section 3, how can we conclude that the degenerate modal behavior is related to the entropy of the set of valid outputs?
> We cannot conclusively conclude that, but entropy is roughly equivalent to there being a larger set of valid answers. As we note, the degenerate mode behavior occurs more frequently for more open-ended prompts and longer source sentences for translation (see Figure 1a), but which we argue is suggestive, but not conclusive that the entropy plays a large role.
>
> ### Question 2
> > Exact search experiments are meaningful, but the result that exact search performs good is not relevant to Subsequent Chapters. Maybe their relevance should be emphasized by experiments in terms of computational cost.
> We open Section 5 by stating “ DFS-related approaches are too expensive to be practical in NLG applications, especially so for the conditioning variables that force high amounts of backtracking.” Does this help with the connection between sections? If not, we can emphasize this further in the writing.
>
> ### Question 3
> > The experiment to prove that ACBS not only benefit from removing empty outputs should be detailed.
>
> The experiment to prove that ACBS not only benefit from removing empty outputs should be detailed
> We in fact controlled for this by making our baseline stronger in empirical comparison! By “restrict our attention” in paragraph 2 of Section 4, we meant that we dropped any examples from our analysis for which beam search (baseline) produces an empty output. This gives the beam search baseline a large advantage, since it isn’t penalized for this failure mode, but ACBS still finds an improved output much more often. As we mentioned above, we have additional results about the quality of generated outputs evaluated by GPT-4 and humans to further support our claims.
>
>
>
> Thank you for the helpful comments and questions, we believe they have helped us significantly improve our manuscript.

---

### Official Review · Reviewer_mRtz · 2023-10-31

**Soundness:** 4 excellent
**Presentation:** 3 good
**Contribution:** 4 excellent
**Rating:** 8
**Confidence:** 4

**Summary:**

This paper argues that one source of so-called text degeneration is contamination of the training data with low-entropy noise, such as empty or nearly empty completions, or partial or total repetitions of the prompt. It substantiates this claim by performing exact MAP decoding, including on LLaMA. It offers a decoding solution, which is to do exact or beam search decoding with constraints on "attributes" like length.

**Strengths:**

Overall, I really like this paper and favor acceptance.

The experiments with exact search provide convincing evidence of the claim about low-entropy strings. These experiments are also valuable because they reveal a new kind of degeneracy (copying the prompt) and they are the first to perform exact decoding on a large language model.

The observation that sampling has the opposite "Achilles' heel" is valuable, although not really the focus of this paper.

The attribute-constrained beam search algorithm is new. It's a nice idea that has essentially the same running time as standard beam search, and seems to work well. It's interesting that the example length-conditioned translations are good summaries of translations.

**Weaknesses:**

The explanation of degeneracy in terms of low-entropy strings is not new, and the authors may not be aware of the following two papers:
Ott, https://arxiv.org/abs/1803.00047
Holtzman, https://arxiv.org/abs/2104.08315

As an alternative to your decoding method, you could use an adaptive beam, using a wider beam for earlier timesteps that gets narrower for later timesteps. Then at timestep $a$, you would get higher-quality outputs with length $a$. I am not sure what schedule you would use for the beam size, but perhaps work by Brian Roark for adaptive beams in CKY parsing is relevant.

The evaluation of the attribute constrained beam search method for LLaMA consists of recording what percent of sentences get a higher reward according to the same reward model used in training, and giving a general subjective impression of sample outputs. I think this is a fairly weak evaluation, and it would be a lot better to elicit quality judgements from other people.

Style / minor points:

There is too much important information in the appendix, especially Algorithm 1, with many references from the text to the appendix.

Everyone has their own writing style, but I feel that there are too many exclamation points for an academic paper. There are even two sentences in a row with exclamation points on page 4, but this is probably just an editing error.

**Questions:**

table 2: how can truncated beam search be better than itself?

table 3: why do the TBS translations seem shorter?

5.3.1 In the example, the source sentence in the prompt is "I love machine learning," but the output translates the source sentence "My eyes are clear." Is that really what happened?

---

> ### Author Response · Authors · 2023-11-21
> **Thank you for your review**
>
> Thank you for your time in reviewing this paper and recognizing the positive aspects of our work! In our updated draft, we provide additional results based on more in depth evaluation and human judgements to further support our claims. Below we respond to your concerns and questions:
>
> ### Additional related work
> We have added brief discussion of both papers you referenced in our updated draft. Ott et al.’s work aligns precisely with the behavior we would expect, although they don’t make the generalization to low-entropy noise in general.
>
> ### Alternative to decoding method
> The suggestion of an adaptive beam size is an interesting one, and should certainly be investigated. Note that if the empty sequence or any other short distractor is the problem, the beam size must start small. On the other hand, for a problem such as repetition of the prompt, a different heuristic may be necessary. These methods (and others such as length-normalization) generally need to be tailored quite closely to the exact degeneracy being combatted.
>
> ### Evaluation with reward model
> We agree with your concern about evaluation based on our reward model. In our updated draft, we report additional metrics that support our claims. In Table 5, we additionally report the results of blinded pairwise comparisons made by GPT-4 and one of the authors. (In a camera-ready version we will add additional human annotators). See also our overall response for other evaluations we have added.
>
> ### Style
> We agree that it is not ideal to put so much content in the Appendix, but we were not able to accommodate it in the main text. We have also cleaned up the writing style in several places per your remarks.
>
> ### Table 2 question
> We realize that the presentation of these results might be confusing. The way to read this is that B=5 and length ratio 0.8, ACBS wins 56.9 % of the time and TBS wins 35.5. % (remaining 7.6% are ties), and for B=20 and length ratio 0.8, ACBS wins 57.8 % of the time and TBS wins 29.1 % (remaining 13.1% are ties). We have  made this clearer in the updated draft.
>
> ### Table 3 question
> The apparent discrepancy in length is due to the tokenizer splitting “It’s” into 3 tokens. For example:
>
> ['▁Andr', 'ada', '▁said', ':', '▁"', 'It', '’', 's', '▁not']
> ['▁Andr', 'ada', '▁said', ':', '▁“', 'It', '▁does', '▁not', '.']
> (Both 9 tokens, excluding EOS)
>
> ### "I love ML" question
> Our original example used the sentence “I love machine learning” in the prompt, but we switched it to “My eyes are clear” once we decided to submit to ICLR.
>
> Thank you once again for your feedback and questions. The additional references in particular have helped quite a bit.

---

### Official Review · Reviewer_YHYo · 2023-11-01

**Soundness:** 2 fair
**Presentation:** 2 fair
**Contribution:** 3 good
**Rating:** 5
**Confidence:** 4

**Summary:**

The authors provide analysis of why text generation models often suffer degenerated distribution mode. They attribute the problem to the contamination in the training data sampled from natural language distribution. They further find that the bad mode problem is alleviated when conditioned on a certain target length. To this end, the authors propose an attribute-conditional beam search algorithm which exhibits superiority compared with truncating methods when target length is given.

**Strengths:**

1. The authors provide a detailed analysis of bad mode problem.
2. The authors propose an attribute-conditional beam search algorithm which exhibits superiority compared with truncating methods when target length is given.
3. The authors conduct experiments on various NLP tasks.

**Weaknesses:**

1. **Analysis less than convincing in supporting the "*bad mode problem*"**.
In Section 2, the authors claim that introducing noise into the data distribution can lead to model degeneration, even when the model is perfectly trained to fit the original data distribution. The authors provide examples to illustrate this concept. However, I found some of these analyses less than convincing. In Section 2.1, the authors argue that "*If one in a billion sequences is replaced with a bad output, MAP on a perfectly trained model should give us one of the bad outputs*". However, this argument relies on the assumption that "*there might be 2^100 possible abstracts for a given scientific paper*". It seems such an assumption never holds true in the case of a real dataset. In contrast, there is only one reference for a source in the typical setting.

2. **Unclear logic between Section 2 & 3.**
In Section 3.2, the authors provide experimental results that "*the occurrence of empty sequences increases with source length*". The authors attribute the empty mode problem to that "*the entropy of valid outputs increases with input length, but the probability of the empty output does not decline enough*". I am confused that this may contradict the analysis in Section 2 that attributes the bad mode to "*low-entropy distractors*".

3. **Experiments are not serious**.
**a.** In Section 3.2.1 & Section 4, the authors conduct qualitative analysis only based on case study, lacking rigorous analysis and discussion.
**b.** In Section 5.2.1, the authors compare their proposed attribute-conditional beam search with truncated beam search when a target length is provided. The comparison is based solely on the log-likelihood of the search results, without considering the evaluation of generated quality, such as BLEU scores. This evaluation seems insufficient, especially considering the background that a language model's mode can lead to degenerated results. Furthermore, there's no comparison between the proposed attribute-conditional beam search and standard beam search (without length truncating, evaluated with both likelihood and BLEU), which appears to be weird and not convincing.
4. **Concerns on the motivation and novelty.**
**a.** The authors proposed a length-conditioned beam search algorithm. However, this seems not very helpful to solving the bad mode problem in LLM, as a pre-determined length may be imprecise and lack the flexibility.
**b.** The novelty of proposed attribute-conditional beam search is limited. Those attribute-conditional sampling methods is well-studied, and the authors only adapt it to the beam search.
5. **The paper has too many typos**.
**a.** Please check the format (should use ICLR 2024)
**b.** Confused paragraph numbering (Section 3.1 & 3.1.1)
**c.** wrong citing format

**Questions:**

Please see the weaknesses above.

---

> ### Author Response · Authors · 2023-11-21
> **Thank you for your review**
>
> Thank you for taking the time to review our paper, providing helpful feedback, and recognizing the detail in which we address the issue of modal sequences in LMs. We have tried to **eliminate the typos** in our newly posted draft and addressed some of your concerns as well.
>
> ### Motivation and novelty
> (See the overall response to reviewers for an explicit list of novel contributions).
>
> We see ACBS as an individual supporting piece of evidence to the overall story, rather than the main novel contribution of the paper. Namely it suggests that our insights and empirical results can be used to drive the design of future MAP-like decoding algorithms. Currently, the decoding algorithms in common use are either sampling based (top-k sampling, top-p sampling, locally typical, FUDGE), or try to maximize some score (Reward-augmented beam search, Monte-Carlo tree search, or the recent methods suggested by Deng and Raffel, 2023, and Mudgal et al., 2023). Here, we point towards another possible class of algorithms: those which     search for sequences of maximal likelihood under a constraint.
>
> The community has largely stopped trying to improve on this class of algorithms, due to what we argue are misconceptions about the reason for degenerate modal sequences. As we argued in Section 2, MAP-like methods have certain desirable properties in terms of controlling noise. We hope that the results we share here will urge the community into further research on this class of algorithms.
>
> ### Concerns about the evaluation of search outputs
>
> In Section 5.2.1, we primarily focused on likelihood because we wanted to emphasize that constrained beam search is not an appropriate method for performing length-constrained MAP finding since our approach can find higher likelihood sequences under the unspecified length-conditional distribution. While it is true that mode of the MT model might be of bad quality, in the sections prior to ACBS, we show that *conditional* modes ameliorate the bad mode problem. Hence, we only reported likelihood scores in our first draft.
>
> To further increase the confidence in this finding, we have added quantitative evaluations throughout our experiments. Please see our summary response for a description of the newly added evaluations.
>
> ### Comments on the example of bad mode problem
> Regarding there only being one reference output in NLG datasets, we are not referring to dataset themselves, but the *population distribution*. We view a dataset as a sample from some population distribution, in line with the common formalism of machine learning. Even in a task like sentence-level machine translation, there are a large number of valid outputs for any given input. So while we only observe one reference, our hope is that our models will learn the population conditional distribution of outputs given inputs. Even for a one word “sentence”, the translations: “Yes”, “That’s right”, “Okay”, “Sure”, “Indeed” and so on may be possible. Which one is exactly correct may depend on things such as the translator’s style, and context which exists outside the sentence level. This set of valid translations explodes in size once we move to full sentences, and is astronomical if one considers a paragraph or full document. In order to reach the 2^100 number for writing scientific abstracts, it just needs to be the case that one makes at least 100 choices with at least 2 options during the process.
>
> Our key point here is that it is necessary that modeling these distributions well also makes the mode of the model more vulnerable to noise.
>
> ### Logic between Section 2 and 3
> The empty sequence is such a low-entropy distractor (more generally you could consider the set of truncated translations of 0-2 tokens to be the set of distractors).
> The number of valid translations for a 30 token input will be enormous, so despite the fact that the probability of the low-entropy set declines (See the newly added Figure 2), the entropy of the good outputs increases too fast. Entropy increasing is the same thing as the average log-likelihood decreasing, so this could equivalently be stated as: The likelihood of a typical good translation decreases with length faster than the likelihood of the distractors decreases with length.
>
> Thank you once again for taking the time to give us detailed feedback on our work.

---

> > ### Comment · Reviewer_YHYo · 2023-11-23
> >
> > Thanks for your reply.
> >
> >
> >
> >
> > According to your comment, you view a dataset as a sample from some population distribution. However, the discussion of the bad mode problem in Section 2 is framed within the context of limited data. I find there is a potential contradiction. In my opinion, with an unlimited amount of data (e.g. the real-world distribution), even if there are some noises, I think the distribution would not experience the bad mode problem. I am willing to have more discussion, however your reply is submitted a little late. In any case, I have decided to raise my score to 5.

---

### Author Response · Authors · 2023-11-21
**Summary response to reviewers, and revisions**

Thank you to all reviewers for taking the time to review our paper and provide constructive feedback. We have made significant changes to our draft based on it, leading to a much better submission. The two most repeated concerns were the lack of quantitative assessments and questions about novelty:

### Quantitative assessment of outputs
The most repeated concern was about quantitative evaluation of the model outputs we share, so we have added additional assessments of output quality throughout:
- Table 2 reports BLEU and BLEURT scores of the modal outputs found by exact search, to provide additional evidence that the length-conditional modes are better.
- Tables 3b and 3c compare beam search and ACBS using BLEU/BLEURT, and perplexity under Llama2-7B. These assessments show that ACBS is indeed outperforming beam search in this case. This adds quantitative evidence to our claim that it finds more fluent outputs which are also assigned higher likelihoods by the MT model, a fact which should be surprising in light of past findings of the likelihood/quality divergence.
- In addition to the comparison of reward from the original draft, Table 5 now reports blinded pairwise assessments by GPT-4 and one of the authors. These assessments both show that ACBS improves over ordinary beam search in terms of quality, and does not simply overfit to the reward function. In a camera-ready version we will increase the number of human evaluators.

### Novelty
The aim of this paper is not to propose a state-of-the-art generation method. Instead, our goal is to argue that MAP-based methods have been prematurely abandoned. Prior work has almost uniformly suggested that the divergence of likelihood and quality in NLG models is due to model error. As we show here, it is easy to construct sequence distributions which show this phenomenon, even when there is **no model error.**

We believe this paper will be valuable to the community, because it urges readers to rethink the assumption that a bad output implies a problem with the model, rather than merely being a counterintuitive property of sequence distributions. Our key claims of novelty are:
- **Exact search results on LLaMA family models**, which are more than an order of magnitude larger than other models which have had exact search results reported. This required modifications to Stahlberg and Byrne’s search method, which we have explained in an appendix.
- **Exact length conditional modes translation and GPT-2 models.** The examples we share contradict the general belief that NLG models do not model extremely high-likelihood sequences well. This information is important for the community to take into account when reasoning about how to make progress on NLG. In our discussions with other NLP researchers, they have typically guessed that conditional modes of NLG models would show degenerate repetition. This is a plausible sense in light of Holtzman et al. (2020), but as we show, the modes turn out to be fluent.
- **A clear description of one cause of the bad mode problem.** No other paper currently makes the point that it *must* be the case that the mode will be low-quality once models are trained well enough on distributions with the right fraction of noise. In particular, that fraction of noise declines exponentially in the entropy-rate of the “clean” distribution.
- **A method for eliciting reasonable instruction-following outputs with no finetuning**: Ours is the first work to report that it is possible to use LLaMA-7B for zero-shot instruction following without any finetuning. This shows that the capability exists in the model already, and the correct type of conditioning can elicit it.

### Summary of revisions
- Added BLEU/BLEURT evaluation of exact search results
- Added BLEU/BLEURT evaluation of ACBS and beam search outputs for exact search
- Added GPT-4/human pairwise comparison of ACBS and beam search outputs for LLaMA-7B instruction following
- Added perplexity evaluation of LLaMA-7B ACBS/beam search outputs to compare fluency
- Added references to suggested related work. In particular, Ott et al. (2018)'s results closely align with the behavior predicted by our argumentation in Section 2
- Improved wording in several places to improve clarity on points raised by reviewers
- Cleaned up formatting and appendices
- Added plot showing that P(EOS) declines with source length, even as the rate of empty modes increases (this was merely stated in the text originally)
- To make room for new results, primarily moved more hyperparameters/details to appendices, shortened conclusion, and improved brevity where possible

---

### Meta-Review · Area_Chair_AKtv · 2023-12-08

**Metareview:**

The paper attributes the degeneracy problem to contamination in the training data and that MAP could still be good if only certain degeneracies are avoided. They proposed a beam searched based generation method (ACBS) that also conditioned attributes like length, which is better compared with truncating methods when target length is given.

3 of 4 reviewers provided negative reviews. The negative reviews generally pointed to how the logic is not clear and the experiments are not rigorous enough to establish the key claims that 1) MAP could still be good with some simple modifications 2) a perfect model would still suffer from degeneracy if the training data contains low entropy noise. There is also some confusion if the generation method or the analysis should be considered the main contribution where the generation method appears to be a simple modification of beam search which reviewers considered insufficient. The authors clarified that the analysis/diagnosis should be considered the main contribution – but the AC agrees with reviewers that it would be reasonable to read ACBS as the main contribution in the paper. All reviews also pointed to roughness in the paper: typos, deferring important details to the appendix, and presentation of the results.

The cause and cure of degeneracy is an important problem to study and the authors are encouraged to focus the paper on establishing their impressive main claims (as in the title) in a revision before going into various side contributions/findings.

**Justification For Why Not Higher Score:**

Clear reviewer majority favoring reject. The main positive review does not point to any fundamental misunderstanding from the negative reviewers or particularly impressive points that warrants overriding the consensus.

**Justification For Why Not Lower Score:**

N/A

---

### Decision · Program_Chairs · 2024-01-16

Reject